# A pair of ascending neurons in the subesophageal zone mediates aversive sensory inputs-evoked backward locomotion in *Drosophila* larvae

**Natsuko Omamiuda-Ishikawa**[1], **Moeka Sakai**[1], **Kazuo Emoto**[1,2]*

**1** Department of Biological Sciences, Graduate School of Science, The University of Tokyo, **2** International Research Center for Neurointelligence (WPI-IRCN), The University of Tokyo

* emoto@bs.s.u-tokyo.ac.jp

**Data Availability Statement:** All relevant data are within the manuscript and its Supporting Information files.

## Abstract

Animals typically avoid unwanted situations with stereotyped escape behavior. For instance, *Drosophila* larvae often escape from aversive stimuli to the head, such as mechanical stimuli and blue light irradiation, by backward locomotion. Responses to these aversive stimuli are mediated by a variety of sensory neurons including mechanosensory class III da (C3da) sensory neurons and blue-light responsive class IV da (C4da) sensory neurons and Bolwig's organ (BO). How these distinct sensory pathways evoke backward locomotion at the circuit level is still incompletely understood. Here we show that a pair of cholinergic neurons in the subesophageal zone, designated AMBs, evoke robust backward locomotion upon optogenetic activation. Anatomical and functional analysis shows that AMBs act upstream of MDNs, the command-like neurons for backward locomotion. Further functional analysis indicates that AMBs preferentially convey aversive blue light information from C4da neurons to MDNs to elicit backward locomotion, whereas aversive information from BO converges on MDNs through AMB-independent pathways. We also found that, unlike in adult flies, MDNs are dispensable for the dead end-evoked backward locomotion in larvae. Our findings thus reveal the neural circuits by which two distinct blue light-sensing pathways converge on the command-like neurons to evoke robust backward locomotion, and suggest that distinct but partially redundant neural circuits including the command-like neurons might be utilized to drive backward locomotion in response to different sensory stimuli as well as in adults and larvae.

## Author summary

In the absence of obstacles, most land animals typically walk forward; aversive cues trigger directional changes that prominently include backward movement. Changes in movement direction are evoked by command neurons in the brain that function on local motor circuits to control direction and timing of muscle movements. *Drosophila* MDNs in the brain act as command-like neurons to evoke the backward movement in adults and

**Funding:** This work is funded by MEXT Grants-in-Aid for Scientific Research on Innovative Areas "Dynamic regulation of brain function by Scrap & Build system" (KAKENHI 16H06456), JSPS (KAKENHI 16H02504), WPI-IRCN, AMED-CREST (JP18gm0610014), JST-CREST, the Strategic Research Program for Brain Sciences, Toray Foundation, Naito Foundation, Takeda Science Foundation, and Uehara Memorial Foundation to K. E; by JSPS fellowship (KAKENHI 19J10763) to N. O.I. The funders had no role in study design, data collection and analysis, decision to publish, or preparation of the manuscript.

larvae, but sensory control of MDN activity is still incompletely understood. Here we identify a pair of ascending neurons, designated as AMBs, that can activate MDNs to elicit backward locomotion in larvae. We present data from functional imaging with optogenetics, anatomical analyses, and behavioral studies that support the idea that AMBs and MDNs are critical components of the neural circuits that transduce aversive stimuli from C4da sensory neurons into backward locomotor output. We further propose that divergent but partially overlapped circuits are recruited to evoke backward locomotion in response to distinct aversive stimuli as well as in adults and larvae. This study paves a way to understand circuit mechanisms of how multiple sensory inputs are coordinated to evoke particular behavior in response to a variety of external cues.

## Introduction

Dynamic locomotion with steering control is critical for animals to avoid unwanted situations [1–3]. Animals generally walk forward but often switch to backward when sensing insurmountable obstacles or potentially dangerous stimuli in their path. As forward locomotion and backward locomotion are mutually exclusive, the selection of walking direction is necessary to prevent injury and escape predation. Thus animals often have dedicated control systems for backward locomotion [4,5]. In the fruit fly *Drosophila melanogaster*, for instance, backward walking relies on small subsets of dedicated neurons including the command-like Moonwalker Descending Neurons (MDNs), two pairs of descending neurons in the brain [6,7]. MDNs are required to walk back from dead ends [6] and to walk away from visual threat [8]. A recent study indicates that the TwoLumps ascending neurons mediate the touch-evoked backward walk through MDN activation in response to touch stimuli on the anterior legs [9]. MDNs also function in *Drosophila* larvae as well as adults to evoke backward locomotion in response to mechanical stimuli on the head [7]. Structural and functional studies indicated that larval MDNs promote backward locomotion through activating the backward specific premotor neurons and simultaneously suppressing forward premotor neurons through distinct postsynaptic partners [7], though sensory control of the larval backward locomotion remains elusive.

In addition to the mechanical stimuli on the head, larvae show backward locomotion when exposed to blue light [10–13], a response that depends on two partially redundant sensory systems, the Bolwig's organ (BO) and the class IV dendritic arborization (C4da) neurons [14]. BO consists of a pair of photoreceptor-containing visual organs located on the larval head which project their axons to the brain [15–17], whereas C4da neurons are multimodal nociceptive neurons that cover the entire larval body wall and project axons to the ventral nerve cord (VNC) [14–17]. Ablation of C4da neurons largely impairs larval responses toward noxious stimuli, whereas optogenetic activation of C4da induces multiple escape behavior including C-shape bending, rolling, and backward locomotion [14,18–24]. Although recent studies have isolated multiple different neurons that act downstream of C4da sensory neurons to evoke rolling behavior [22,23,25,26], little is known about the circuits downstream of C4da neurons that control backward locomotion. Furthermore, how BO and C4da sensory circuits are integrated to evoke robust backward locomotion upon blue light stimuli remains largely unknown.

In this study, through a non-biased optogenetic screen, we identify a pair of neurons in the subesophageal zone (SEZ), designated as AMBs, that induce repetitive and intensive backward locomotion in larvae. Our structural and functional analysis suggests that AMBs function upstream of MDNs in order to elicit backward locomotion. Further behavioral and functional

analysis shows that AMBs preferentially relay aversive stimuli from C4da neurons, but not from BO, to MDNs to trigger backward locomotion. We also found that, unlike in adult flies, MDNs are dispensable for the dead end-evoked backward locomotion in larvae. Our findings thus uncover neural circuits underlying blue light-triggered backward locomotion and suggest that distinct but partially redundant circuits mediate backward locomotion in response to different stimuli.

## Results

### A pair of ascending cholinergic neurons in SEZ can trigger backward locomotion

In order to systematically identify neurons involved in control of a specific type of escape behavior, backward locomotion, we conducted an optogenetic screen using the Janelia GAL4 collection together with *UAS-CsChrimson* [27,28] (see Materials and Methods for details). From 783 GAL4 lines screened, we identified three GAL4 lines whose activation triggered robust backward locomotion in third instar larvae (Fig 1A, S1 Fig, S2 Fig and S1 Movie). Prior studies identified the Moonwalker Descending Neurons (MDNs) as command neurons that trigger backward locomotion upon activation [7], therefore we examined whether any of our GAL4 lines label MDNs. We found that one of the three GAL4 lines, *R73F04-GAL4*, appeared to label MDNs based on the cell morphology and location in the brain. Indeed, dual labeling with *R73F04-LexA* and *SS01613-GAL4* that labels MDNs [7] confirmed that the two pairs of descending neurons labeled by *R73F04-GAL4* are identical to MDNs (S1 Fig).

Since the other two GAL4 lines, *R60F09-GAL4* and *R73D06-GAL4*, did not label MDNs, we reasoned that these lines would facilitate identification of new neurons involved in backward locomotion. To genetically define these neurons, we employed combinatorial expression of a variety of GAL80 and FLP drivers together with *R60F09-GAL4* [29–31]. Using these approaches, we subdivided *R60F09-GAL4*-positive neurons into distinct populations (Fig 1A–1C: see S1 Table for genotypes of each GAL4 lines and S2 Table for all the numerical data). Optogenetic activation of two of these populations labeled by *R60F09-ACh-GAL4*, but not neurons labeled by *R60F09-Brain-GAL4*, significantly increased the number of backward waves (Fig 1E and 1F). Since a pair of neurons in the subesophageal zone (SEZ) was observed in the *R60F09-ACh-GAL4* population, but not in the *R60F09-Brain-GAL4* population (Fig 1B arrowheads), we focused our later studies on these neurons as potential candidates responsible for backward locomotion, and designated these neurons as Ascending Moonwalker-like Backward neurons (AMBs).

To further examine whether AMBs are responsible for backward locomotion, we next checked whether AMBs were labeled by *R73D06-GAL4* as well as *R60F09-GAL4*. Dual labeling with *R60F09-LexA* and *R73D06-GAL4* indicated that AMBs in the SEZ are the only cells in the larval CNS co-expressing both drivers (S2 Fig). We took advantage of this restricted intersection to test the contribution of AMBs to backward locomotion induced by optogenetic stimulation of *R73D06-GAL4*-expressing cells. In larvae expressing *LexAop-GAL80* with *R60F09-LexA* in addition to *UAS-CsChrimson* with *R73D06-GAL4*, backward responses as well as CsChrimson expression in AMBs were no longer observed on light application (S2 Fig), further indicating that AMB activation triggers backward locomotion. In addition to backward locomotion, optogenetic activation of *R-73D06-GAL4* neurons evoked rolling behavior (S2 Fig). Unlike backward locomotion, rolling behavior was unaffected by *R60F09-GAL80* (S2 Fig), suggesting that rolling behavior evoked by optogenetic activation of *R-73D06-GAL4* neurons is mediated by other neurons rather than AMBs. Finally, we generated a splitGAL4 by combining *R60F09-GAL4DBD* and *11E07-p65AD*, and confirmed that the

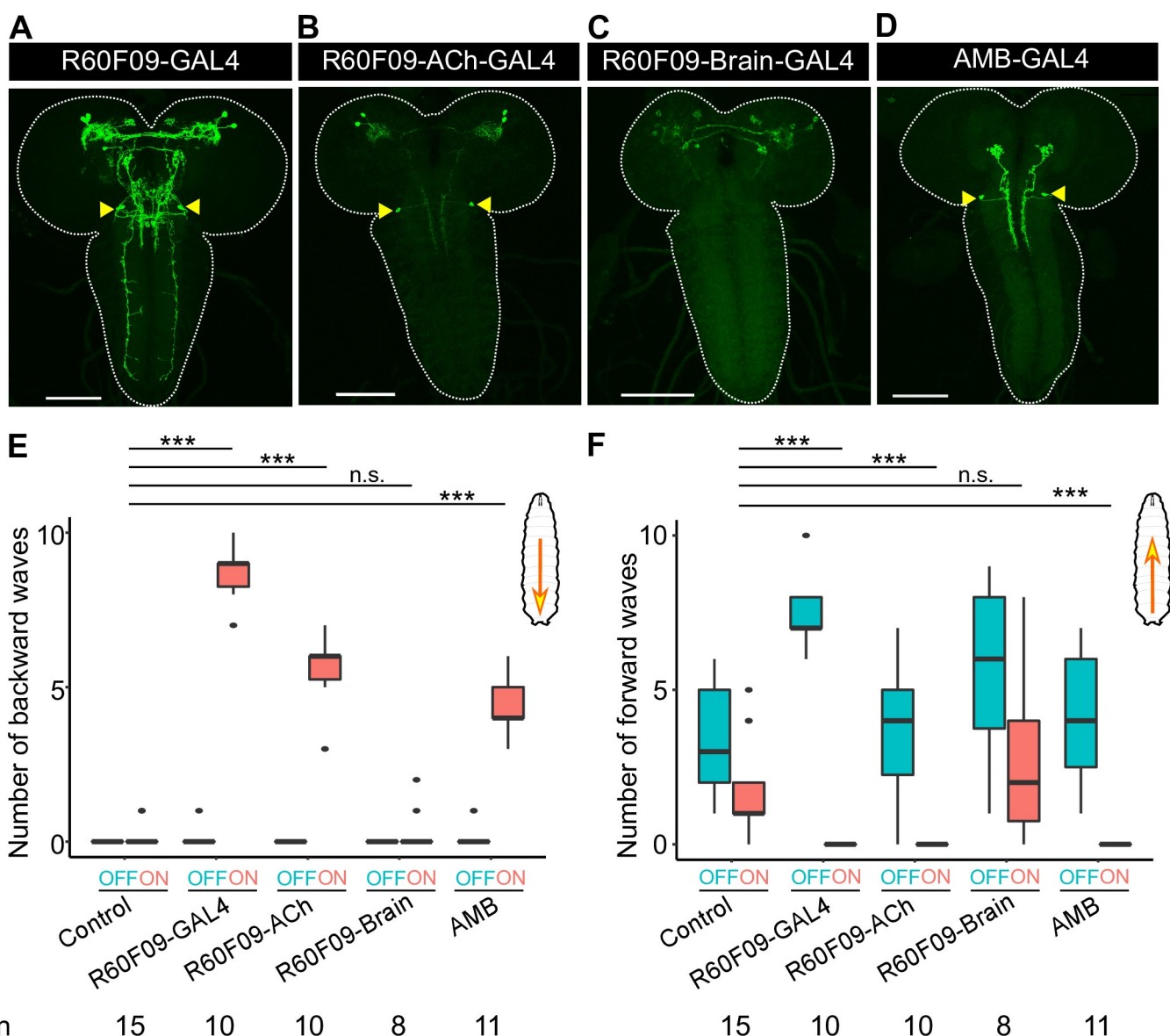

**Fig 1. AMBs are a pair of neurons in SEZ that can induce backward locomotion upon activation.** (A-D) Expression patterns of GAL4 lines. The yellow arrowheads indicate the somas of AMBs. Scale bar, 100 μm. Genotypes of each GAL line are shown in S1 Table. (E, F) The number of backward/forward waves in 10 seconds before (OFF) or during (ON) optogenetic activation with CsChrimson. Genotypes: *w; UAS-CsChrimson*/+; +/+ (Control); other four lines drive CsChrimson by GAL4 lines shown in (A-D). Control, n = 15; *R60F09-GAL4*, n = 10; *R60F09-ACh-GAL4*, n = 10; *R60F09-Brain-GAL4*, n = 8; *AMB-GAL4*, n = 19. In the boxplot, the width of the box represents the interquartile range and the dot represent outlier in the graph. The whiskers extend to the data point which is less than 1.5 times the length of the box away from the box, and the dot represent outlier. We assessed statistical significance using the Wilcoxon rank sum test and corrected for multiple comparisons using the Holm method. ***p<0.001.

split-GAL4 specifically labeled AMBs in the larval brain (Fig 1D) (hereafter, designated as *AMB-GAL4*). Optogenetic stimulation of *AMB-GAL4*-expressing cells induced significant backward locomotion (Fig 1E and 1F). Taken together, we concluded that AMBs trigger backward locomotion upon optogenetic activation.

To further characterize AMBs, we expressed the dendritic marker Denmark [32] and the presynaptic marker BrpD3::mCherry [33] in AMBs. Denmark labeled the somas and neurites from the SEZ to the T2 segment, whereas BrpD3::mCherry localized in the neurites projecting

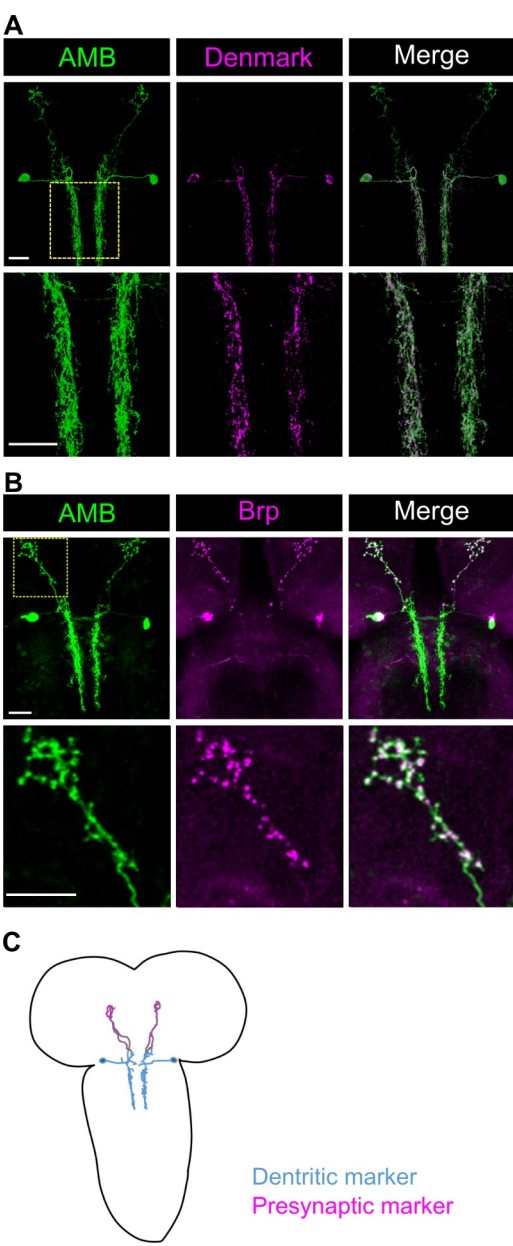

**Fig 2. Anatomical characterizations of AMBs.** (A, B) The dendritic marker Denmark (A) or the presynaptic marker Brp::mCherry (B) were co-expressed with the membrane-localized GFP in AMBs. The area indicated by yellow dot square was magnified in the lower panels. Genotypes: *w*; *UAS-mCD8GFP/VGlut-GAL80 [MI04979]*; *R60F09-GAL4/ UAS-Denmark* (A); *w*; *UAS-mCD8GFP/VGlut-GAL80 [MI04979]*; *R60F09-GAL4/UAS-brpD3::mCherry* (B). Scale bar, 20 μm. (C) A schematic anatomy of AMBs. The color of neurites indicate the soma and the dendritic arbors (blue) and presynaptic site in the axonal processes (magenta), respectively.

to the brain (Fig 2A and 2B), indicating that AMBs are ascending neurons that have dendritic arborizations in the SEZ to the T2 segment and extend axonal projections to the brain (Fig 2C). We also analyzed neurotransmitter expression in AMBs and found that AMBs were immunoreactive to the choline acetyltransferase (ChAT), but not to the vesicular glutamate transporter (VGlut) or GABA (Fig 3 and S3 Fig), indicating that AMBs are cholinergic neurons.

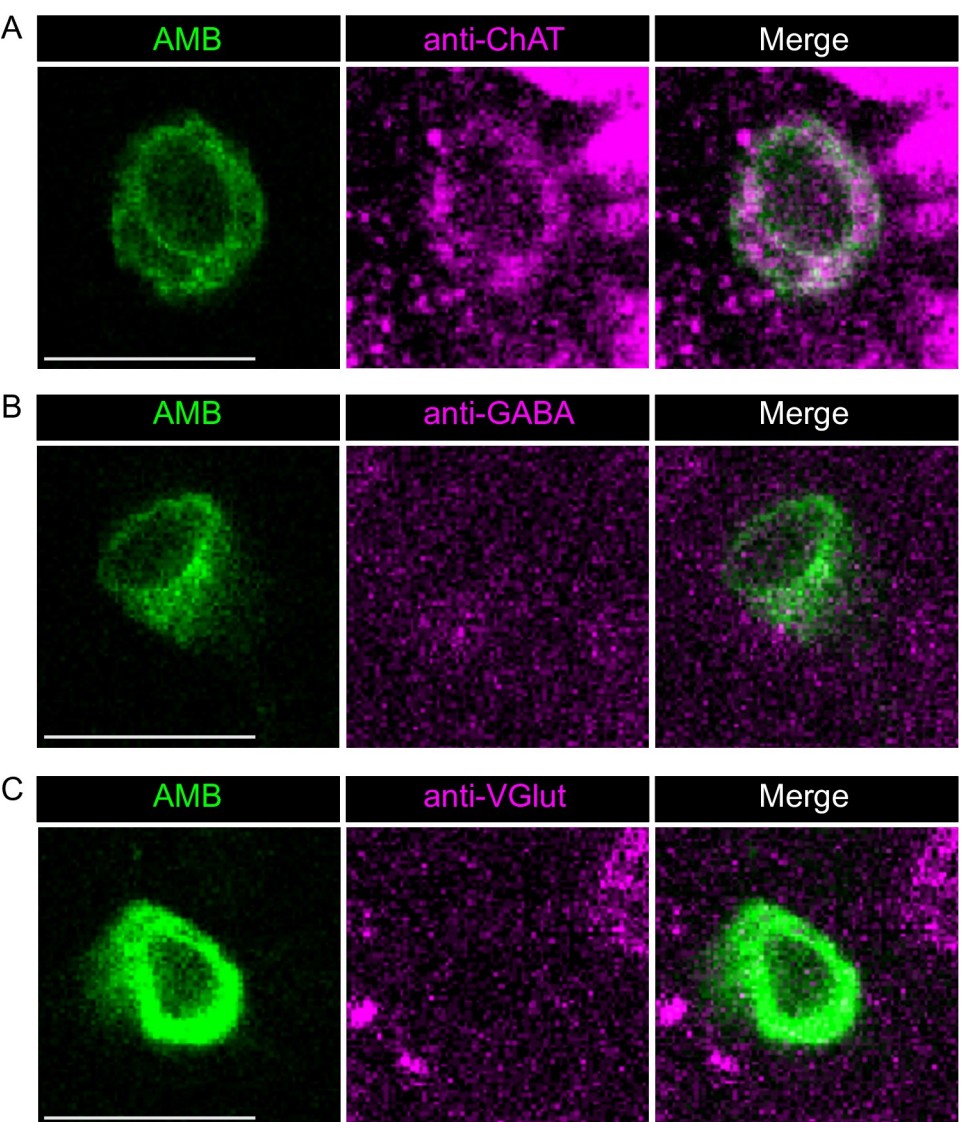

**Fig 3. AMBs are immunoreactive to ChAT, but not to VGlut and GABA.** AMBs expressing membrane-localized GFP were immunostained by antibodies against three different neurotransmitter markers: (A) ChAT, (B) GABA, and (C) VGlut. Scale bars, 10 μm.

## AMBs function upstream of MDNs

Given that activation of AMBs and MDNs triggers similar repetitive backward locomotion (S4 Fig), we reasoned that AMBs and MDNs might function in the same neuronal circuit to control backward locomotion. To test this possibility, we dual-labeled AMBs and MDNs using the GAL4/UAS and the LexA/LexAop binary expression systems, and found that AMB axons are closely apposed to MDN dendrites in the brain (Fig 4), implying that AMBs might function upstream of MDNs. To further test this possibility, we monitored synaptic GRASP (GFP reconstitution across synaptic partners) signal using the t-GRASP system, which relies on split-GFP fragments targeted to each side of the synapse [34,35]. Using this assay, we observed GFP signals in MDN dendrites at sites of AMB axon contacts in larvae expressing pre- and post-t-GRASP fragments in AMBs and MDNs, respectively (Fig 5B). In contrast, we detected

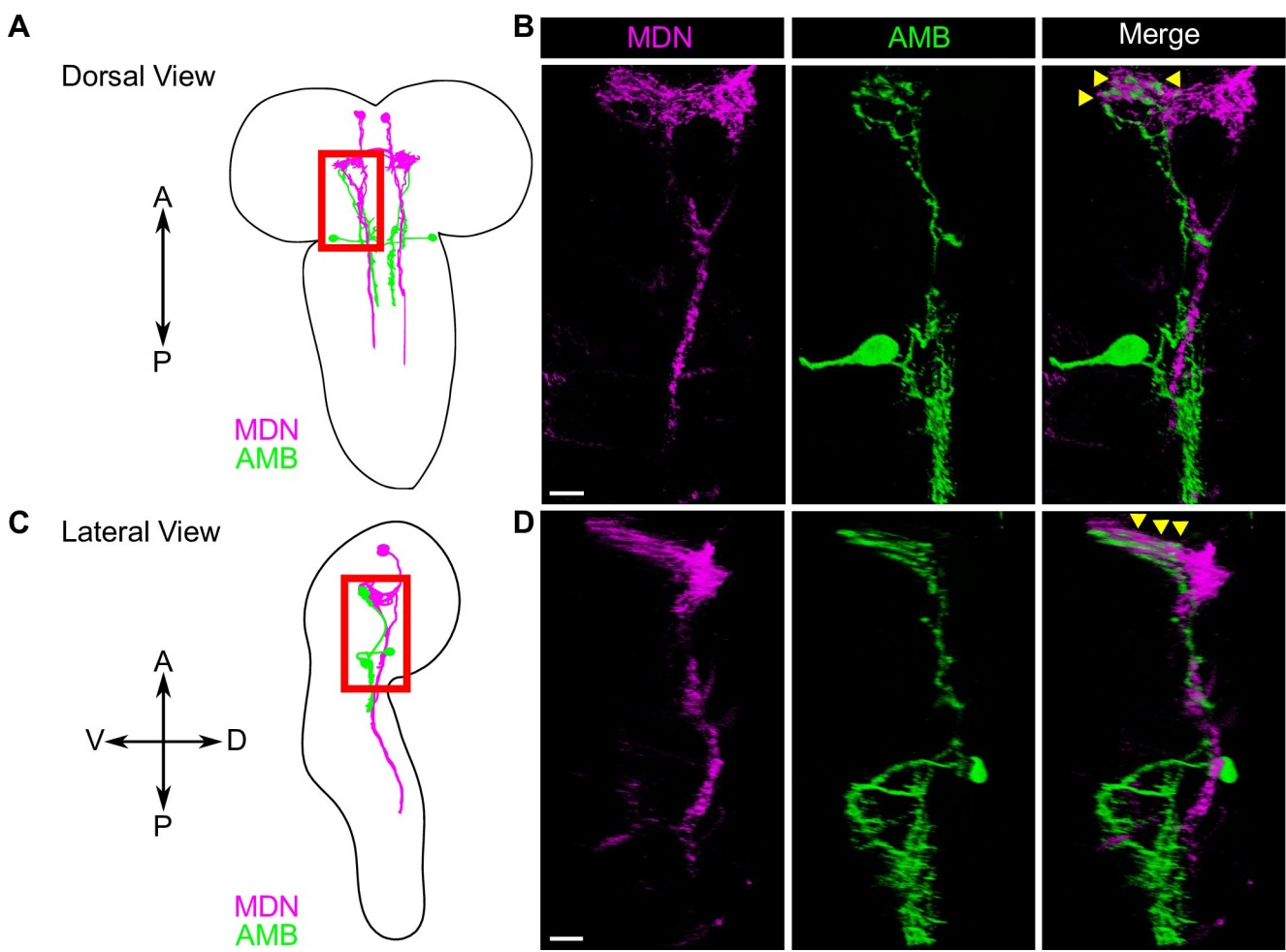

**Fig 4. AMB axons are apposed to MDN dendrites in the larval brain.** (A-D) A schematic view of AMBs and MDNs from the dorsal (A) and from lateral side (C). The area with red boxes in (A) and (C) were shown in (B) and (D), respectively. Dual-labeling of AMBs (green) and MDNs (magenta). AMBs and MDNs were co-labeled with membrane-localized RFP and membrane-localized GFP, respectively. Genotypes: *w; R60F09-LexA, tsh-GAL80/LexAop-rCD2RFP; R73F04-GAL4, Gad1-2A-GAL80/UAS-mCD8GFP*. Scale bars, 10 μm.

no obvious GFP signal in larvae expressing only pre-t-GRASP fragments in AMBs (Fig 5D). These data suggest that AMB axons form contacts with MDN dendrites.

Next, to investigate the functional connection between AMBs and MDNs, we performed calcium ($Ca^{2+}$) imaging of MDNs upon AMB activation. To this end, we expressed CsChrimson in AMBs and the calcium sensor GCaMP6m in MDNs, and we monitored $Ca^{2+}$ responses at MDN axons following optogenetic AMB stimulation (Fig 6A). In a semi-intact preparation of the larval CNS [22], red light application significantly increased GCaMP6m signal intensity in MDN axons (Fig 6; 4.97% ± 0.74% elevation of $\Delta F/F_0$ on average, n = 34). By contrast, the GCaMP6m intensity was largely unchanged in MDN axons of larvae not expressing CsChrimson in AMBs (Fig 6; 0.09% ± 0.60% elevation of $\Delta F/F_0$ on average, n = 24). These data indicate that AMB activation induces $Ca^{2+}$ elevation in MDNs, supporting the idea that AMBs are functionally coupled to MDNs.

Finally, we asked whether MDNs might function downstream of AMBs to evoke backward locomotion. To test this possibility, we simultaneously activated AMBs with CsChrimson and silenced MDNs via expression of the tetanus neurotoxin light chain (TNT) (Fig 7A) [36].

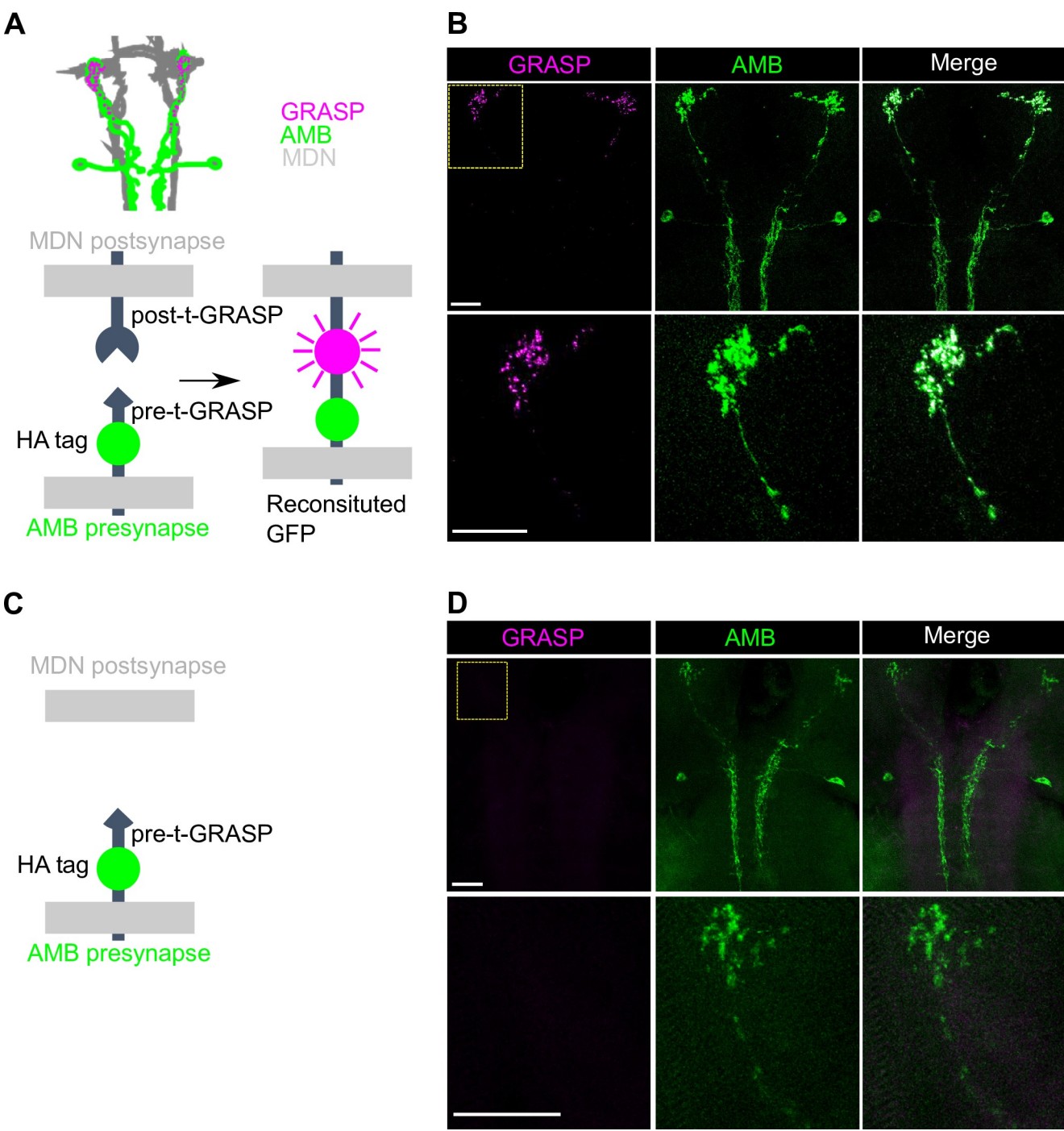

**Fig 5. GRASP analysis of physical contacts between AMBs and MDNs.** (A) A schematic view of t-GRASP between AMBs and MDNs. spGFPs located in each site of the synapse are reconstituted only when two membranes are close enough to make contacts. (B) GRASP signals between AMBs and MDNs were observed the area where AMB axons were co-localized with MND dendrites in the brain. (C) A schematic view of t-GRASP between AMBs and MDNs without expression of the post-t-GRASP fragment in MDNs. (D) No detectable GRASP signals were observed in the brain without expression of the post-t-GRASP fragment in MDNs. The yellow dot square in the upper row is magnified in the lower row. AMBs are labeled by anti-HA. Genotypes: *w*; *R60F09-LexA/+*; *R73F04-GAL4, Gad1-2A-GAL80/UAS-post-t-GRASP, pre-t-GRASP*. Scale bars, 20 μm.

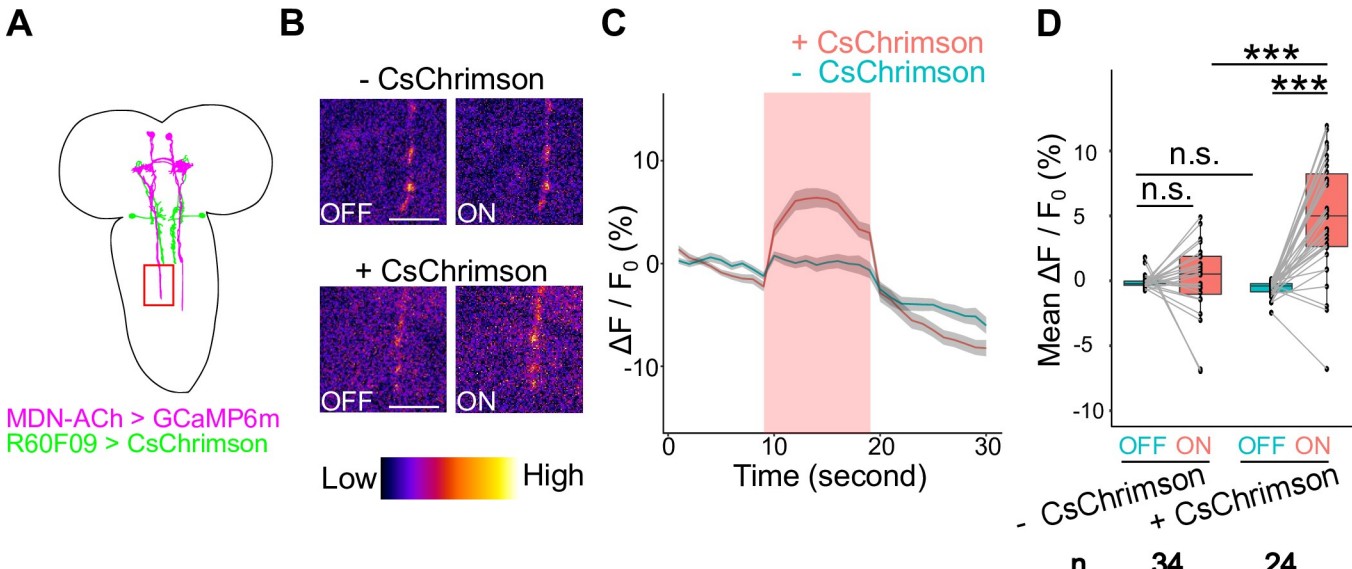

**Fig 6. Optogenetic AMB activation causes Ca²⁺ responses in MDNs.** (A) A schematic view of AMBs and MDNs. The red box indicates the area observed in Ca²⁺ imaging. (B) Ca²⁺ imaging of MDNs upon optogenetic activation of AMBs in with (lower panels) or without CsChrimson (upper panels) conditions. Here are shown representative images of relative Ca²⁺ levels 5 seconds before (OFF) and after (ON) light application. Scale bars, 10μm. (C) Time series of calcium responses in MDN axons upon optogenetic AMB activation. Red light was applied during the period indicated by the red band. Larvae harboring CsChrimson, n = 24; Larvae without CsChrimson, n = 34; Data are shown as the mean ± SEM. Genotypes: *w; UAS-GCaMP6m, tsh-GAL80/R60F09-LexA; R73F04-GAL4, Gad1-2A-GAL80/LexAop-CsChrimson* (+*CsChrimson*); *w; UAS-GCaMP6m, tsh-GAL80/+; R73F04-GAL4, Gad1-2A-GAL80/LexAop-CsChrimson* (-*CsChrimson*). (D) Average of MDN ΔF/F₀ values in 10 seconds before (OFF) or during (ON) optogenetic activation. In the boxplot, the width of the box represents the interquartile range. The whiskers extend to the data point which is less than 1.5 times the length of the box away from the box, and the dot represent outlier. We assessed statistical significance by Welch's t test with Holm method. ***p < 0.001.

Silencing MDNs significantly decreased the number of backward waves triggered by AMB activation compared with control (Fig 7D; effector control, 3 backward waves in the median, n = 17; *MDN-ACh-GAL4* silencing, 1 backward wave in the median, n = 15), whereas silencing MDNs did not significantly affect the number of forward waves before AMB activation (Fig 7E; effector control, 3 forward waves in the median, n = 17; *MDN-ACh-GAL4* silencing, 2 forward waves in the median, n = 15). Furthermore, behavioral ethograms of AMB activation (*R60F09>CsChrimson*) and AMB activation with MDN silencing (*R60F09>CsChrimson, MDN-ACh>TNT*) showed that silencing MDN preferentially attenuated backward locomotion evoked by optogenetic AMB activation, but not other behaviors including forward locomotion and bending (Fig 7F and 7G; *R60F09>CsChrimson*, percentage of time spent on bending 10 seconds after light stimulation onset 24.0% in the median, stop 21.0%, n = 17; *R60F09>CsChrimson, MDN-ACh>TNT*; bending 37.8%, stop 28.5%, n = 14). Thus, AMB activation triggers backward locomotion at least in part through MDN activity. Taken together, these data indicate that AMBs function upstream of MDNs to evoke backward locomotion.

## AMBs preferentially convey aversive information from C4da sensory neurons to MDNs to evoke backward locomotion

Given that AMBs are sufficient to trigger backward locomotion, we next performed silencing experiments to test whether AMBs are required for backward locomotion in response to sensory stimuli. Previous studies indicate that two different stimuli can evoke backward locomotion in larvae: mechanical stimuli [7] and blue light irradiation [13] on the head. We first confirmed that gentle mechanical touch using an eyelash probe induced significant backward locomotion in larvae (Fig 8 and S2 Movie). Next, we examined whether eyelash-triggered

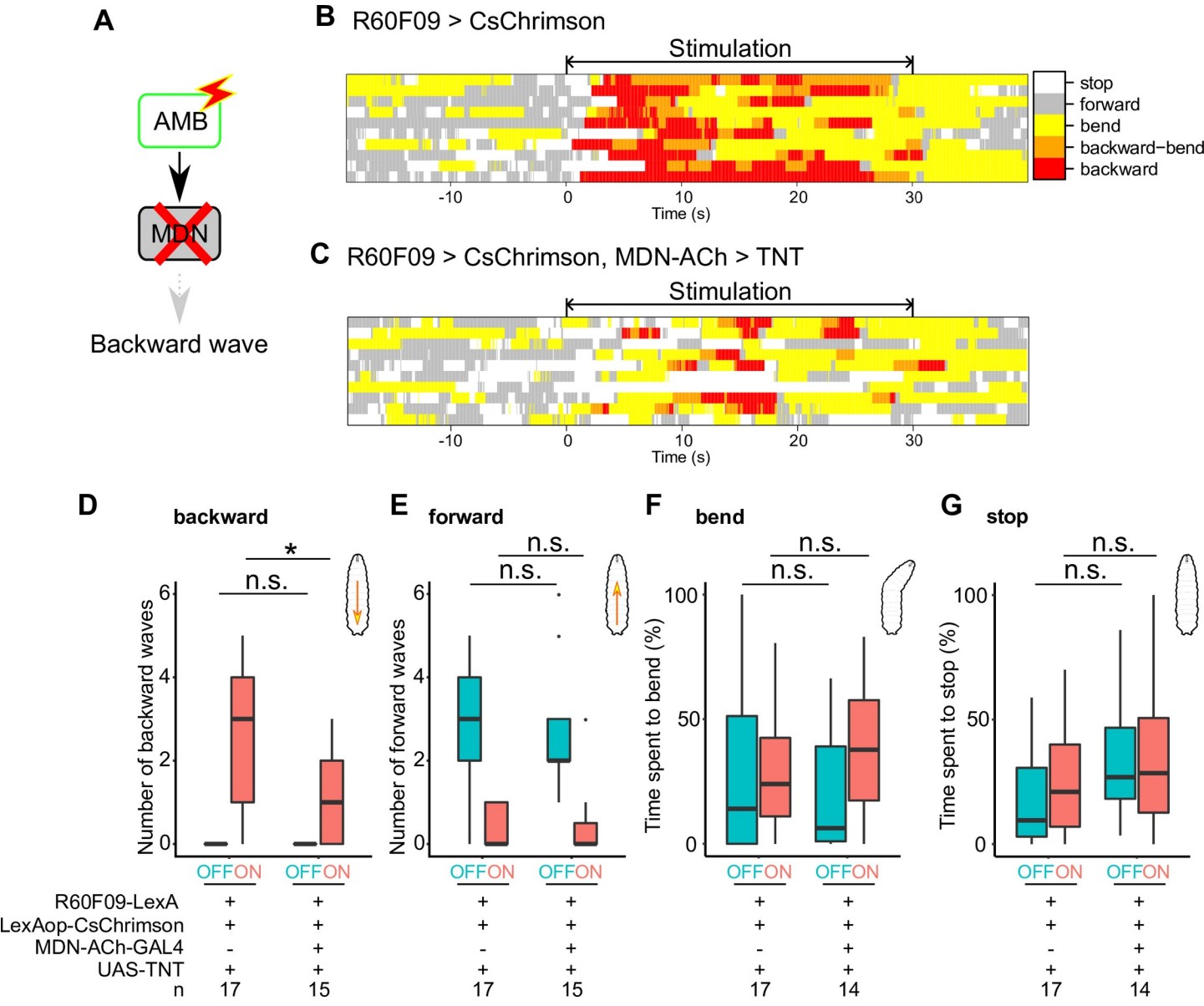

**Fig 7. AMBs require MDNs to evoke backward locomotion.** (A) A schematic view of optogenetic AMB activation with silencing MDNs. (B, C) Representative behavior ethograms upon optogenetic stimulation of AMB neurons. An animal expressing CsChrimson in AMBs was subjected to optogenetic activation for 30 seconds. Behavior events are color-coded: forward movement (grey), stop (white), bending (yellow), bending with backward locomotion (orange) and backward locomotion (red). Representative data from 10 different animals are shown for each genotype. (D-G) The number of backward/forward waves or percentage of time spent in a behavioral mode in 10 seconds before (OFF) and during (ON) optogenetic AMB activation with CsChrimson while silencing MDNs. In the boxplot, the width of the box represents the interquartile range. The whiskers extend to the data point which is less than 1.5 times the length of the box away from the box, and the dot represent outlier. We assessed the statistical significance by the Wilcoxon rank-sum test with the Holm method. *p < 0.05.

backward responses are predominantly mediated by the gentle touch-responsive class III da (C3da) mechanosensory neurons [13]. We genetically ablated C3da neurons by expressing the pro-apoptotic genes *reaper* (*rpr*) and *head involution defective* (*hid*) [37–39] using the C3da neuron-labeling GAL4 drivers *GAL4*[19-12] and *NompC-GAL4* [40] and found that eyelash-triggered backward responses were largely abolished by ablation of C3da neurons (Fig 8A; Effector control, 1 in the median, n = 15; *GAL4*[19-12] ablation, 0 in the median, n = 30; *NompC-GAL4* ablation, 0 in the median, n = 30), but not by ablation of C4da neurons (Fig 8A; *ppk-GAL4*

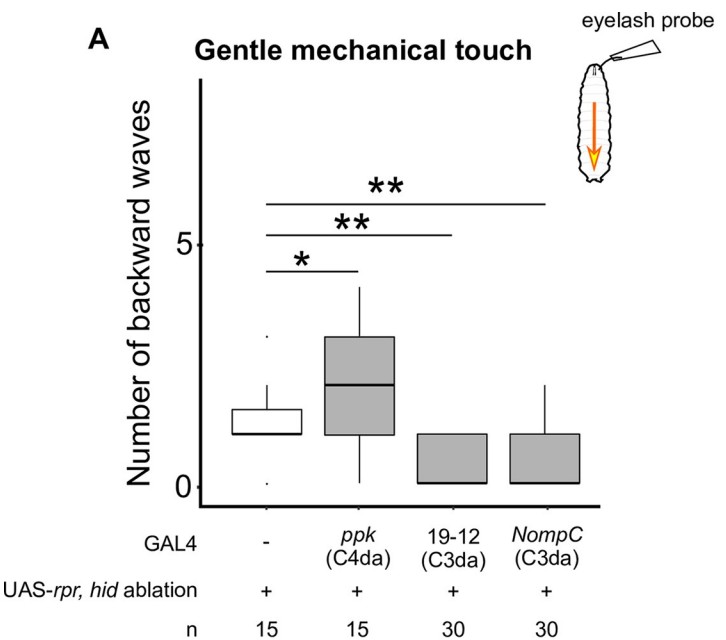

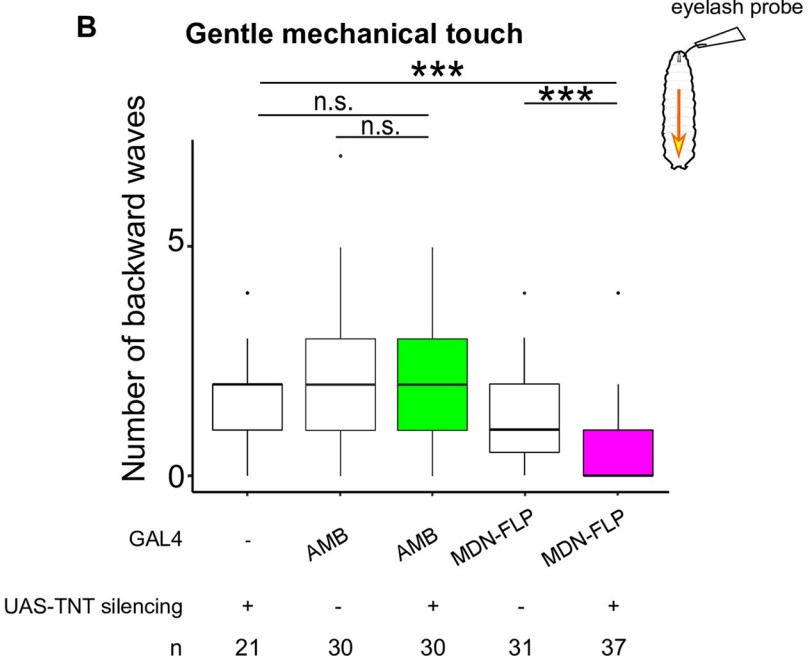

**Fig 8. Gentle touch-induced backward waves are mediated by C3da and require MDN activity** The number of backward waves in response to gentle touch with ablation of sensory neurons (A) or silencing AMBs or MDNs (B). Gentle touch was applied four times with 15 seconds intervals. Genotypes are shown in S1 Table. In the boxplot, the width of the box represents the interquartile range. The whiskers extend to the data point, which is less than 1.5 times the length of the box away from the box, and the dot represent the outlier. We assessed the statistical significance by the Wilcoxon rank-sum test with the Holm method. * p < 0.05, ** p < 0.01, *** p < 0.001.

ablation, 2 in the median, n = 15). Next, we tested the requirement of AMBs in the gentle touch-triggered backward responses. We silenced AMBs by expressing TNT using *AMB-GAL4* and found no significant defects in gentle touch-triggered backward responses compared to

control larvae (Fig 8B and S2 Movie; Effector control, 2 backward wave in the median, n = 21; *AMB-GAL4* control, 2 backward wave in the median, n = 30; *AMB-GAL4* silencing, 2 backward waves in the median, n = 30). By contrast, consistent with a previous report [7], silencing MDNs attenuated the backward response (Fig 8B and S2 Movie; *MDN-FLP-GAL4* control, 1 backward wave in the median, n = 31; *MDN-FLP-GAL4* silencing, 0 backward waves in the median, n = 37). These data suggest that AMBs are dispensable for the gentle touch-induced backward response.

Next, we tested whether AMBs and MDNs are required for blue light-triggered backward responses, as the requirement of AMBs as well as MDNs was not yet examined in this behavioral context. Previous studies suggested that two distinct sensory systems, the Bolwig's organ (BO) and the class IV dendritic arborization (C4da) neurons, mediate blue light-sensing in larvae [11,14]. To examine the relative contribution of BO and C4da neurons to the blue light-induced backward response, we ablated each of them alone or in combination via expression of *rpr* and *hid*. We found that ablation of both BO and C4da neurons caused a significant reduction in the probability of animals exhibiting backward waves compared to the effector control, whereas the probability of animals exhibiting backward waves was unaffected by ablation of either BO or C4da neurons alone (Fig 9A; Effector control, 56.7%, n = 30; *ppk-GAL4* ablation, 53.3%, n = 30; *GMR-GAL4* ablation, 33.3%, n = 30; *ppk-GAL4* + *GMR-GAL4* ablation, 7.4%, n = 27). We also assessed whether C3da neurons could contribute to blue light-induced backward response by ablating C3da neurons or combination of C3da neurons and BO. We found no significant difference in the probability of backward crawling by ablation of either C3da neurons alone or combination of C3da neurons and BO (*GAL4^{19-12}* ablation, 30%, n = 30; *NompC-GAL4* ablation, 45.2%, n = 31; *GAL4^{19-12}* + *GMR-GAL4* ablation, 26.7%, n = 30; *NompC-GAL4* + *GMR-GAL4* ablation, 53.3%, n = 30). Thus, BO and C4da pathways, but not C3da, function redundantly to trigger blue light-triggered backward locomotion.

We next assayed the requirement for neuronal activity of AMBs and MDNs in blue light-triggered backward locomotion. To this end, we selectively expressed TNT in AMBs or MDNs and then induced blue light-induced locomotion. Compared with effector or GAL4 controls, silencing MDNs significantly decreased the probability of animals exhibiting backward responses in response to blue light stimulation (Fig 9B and S3 Movie; Effector control, 45.7%, n = 35; *MDN-FLP-GAL4* control, 38.1%, n = 21; *MDN-FLP-GAL4* silencing, 6.7%, n = 30). By contrast, silencing AMBs had no measurable effect on the probability of animals exhibiting backward responses (Fig 9B; *AMB-GAL4* control, 33.3%, n = 30; *AMB-GAL4* silencing, 43.3%, n = 30). These data indicate that MDNs, but not AMBs, are required to evoke backward locomotion upon blue light irradiation.

Given that BO and C4da pathways function redundantly to evoke backward locomotion upon blue light irradiation and that AMBs function upstream of MDNs, we reasoned that AMBs might preferentially relay information from either BO or C4da pathway to MDNs. To test this hypothesis, we attempted to activate either BO or C4da neurons optogenetically while simultaneously silencing either MDNs or AMBs with TNT. We first test whether optogenetic activation of either C4da neurons or BO could induce backward locomotion using *R27H06-LexA* and *Rh6-LexA* to drive specific expression of CsChrimson in C4da and BO neurons, respectively [22,41]. Optogenetic activation of C4da neurons induced multiple escape behaviors including bending, rolling, and backward locomotion (bending 100% of observed larvae, rolling 55.0%, backward locomotion 35.0%, n = 40). During optogenetic C4da activation for 30sec, rolling tended to be evoked in the earlier time (0–5 seconds) whereas backward locomotion was observed in the later time (5–30 seconds). Similarly, optogenetic activation of BO evoked bending and backward locomotion, but not rolling (bending 100% of observed larvae, backward locomotion 32.5%, n = 40).

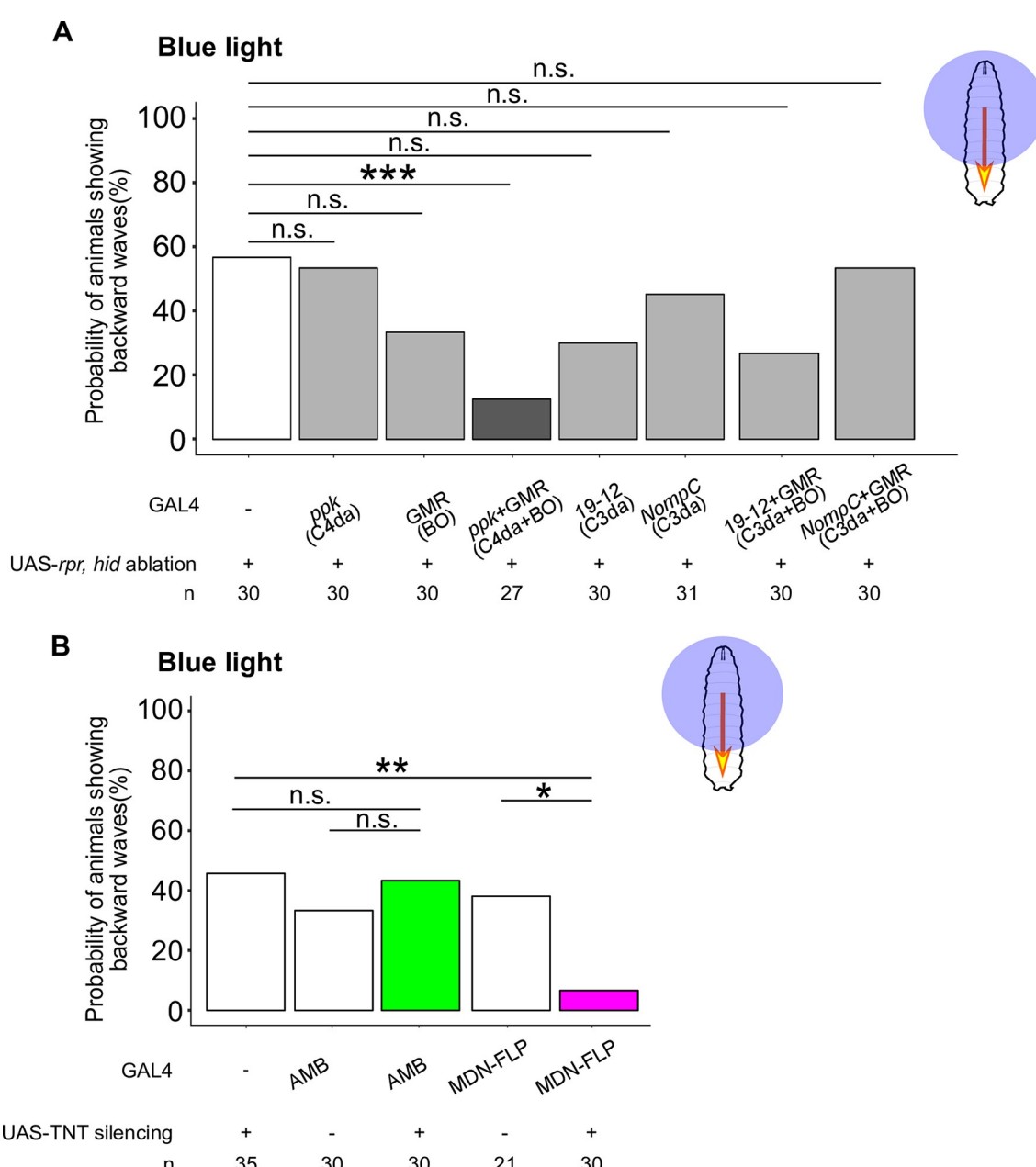

**Fig 9. Blue light-induced backward waves are mediated by C4da and BO pathways and require MDN activity.** The probability of animals exhibiting backward waves upon blue light application with ablation of sensory neurons (A), or silencing AMBs or MDNs (B). Blue light was applied for 5 seconds. Genotypes are shown in S1 Table. We assessed the statistical significance by Fisher's exact test with the Holm method. * p < 0.05, ** p < 0.01.

Next, we assayed effects of MDN or AMB silencing on optogenetically induced backward locomotion. Silencing MDNs significantly decreased the probability of animals exhibiting backward locomotion induced by either C4da or BO activation (Fig 10; C4da activation with *MDN-FLP-GAL4* silencing, 5.0%, n = 40; BO activation with *MDN-FLP-GAL4* silencing, 7.5%, n = 40), consistent with a previous report that MDNs are command-like neurons for mechanosensory-evoked backward responses [7]. Silencing AMBs using two different GAL4 drivers, *R60F09-ACh-GAL4* and *R73D06-GAL4* (S2 Fig), similarly decreased the probability of animals

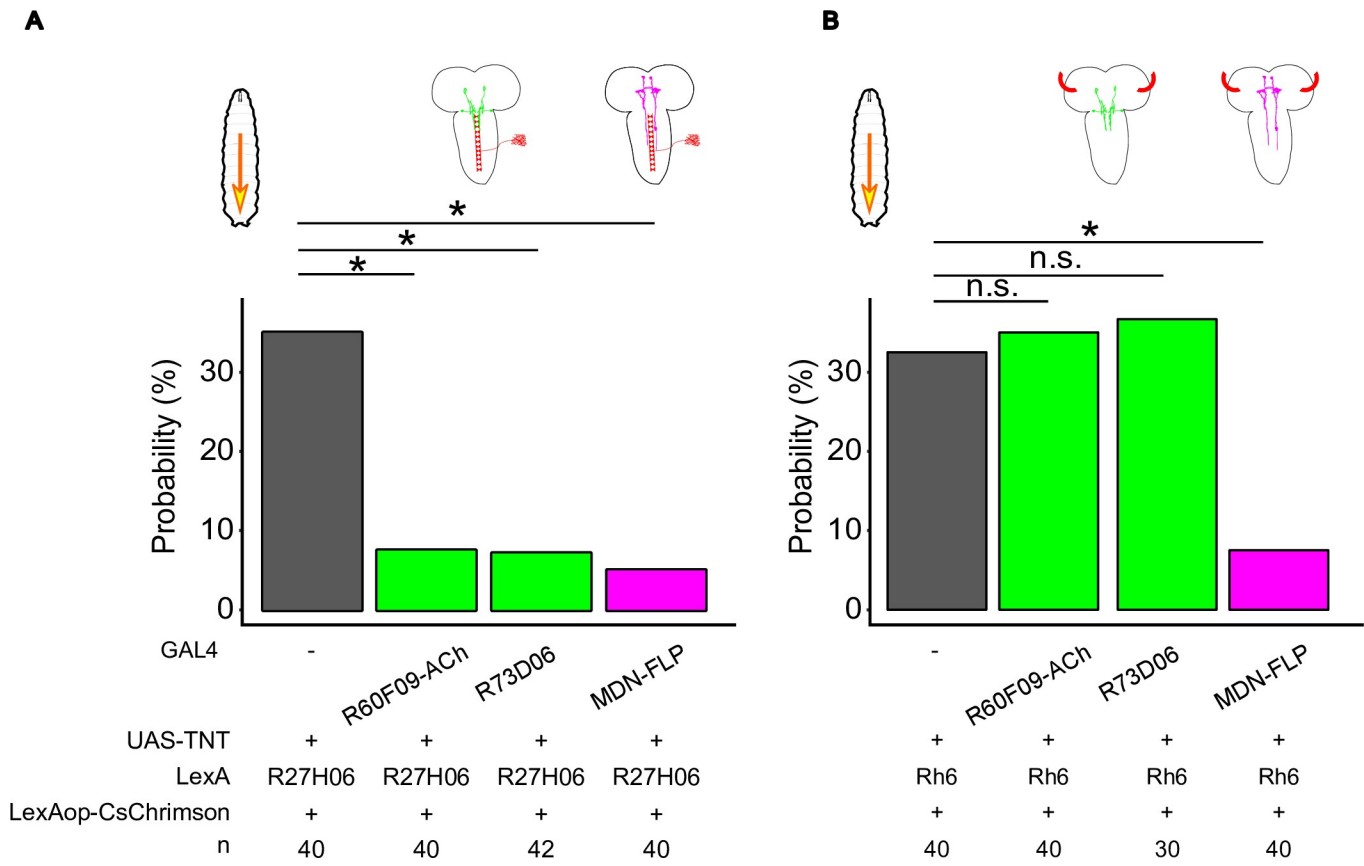

**Fig 10. C4da neurons require AMBs and MDNs to evoke backward locomotion.** The probability of animals exhibiting backward waves during optogenetic activation of C4da neurons (A) or BO (B) for 30 seconds. Genotypes shown in SI Table. We evaluated statistical significance by Fisher's exact test with the Holm method. $^{**}p<0.01$, $^{*}p < 0.05$.

exhibiting backward waves in response to activation of C4da neurons (Fig 10; C4da activation with *R60F09-ACh-GAL4* silencing, 7.5%, n = 40; C4da activation with *R73D06-GAL4* silencing, 7.1%, n = 42). However, backward responses induced by BO activation were largely unaffected by silencing AMBs (Fig 10; BO activation with *R60F09-ACh-GAL4* silencing, 35.0%, n = 40; BO activation with *R73D06-GAL4* silencing, 36.7%, n = 30). These data together suggest that AMBs preferentially mediate C4da-induced backward locomotion.

Finally, in order to confirm the functional connection between AMBs and sensory organs, we asked whether activation of either BO or C4da neurons could trigger $Ca^{2+}$ responses in AMBs. We expressed GCaMP6m in AMBs and CsChrimson in sensory neurons, and monitored $Ca^{2+}$ responses in AMBs following BO or C4da activation. Optogenetic activation of C4da neurons, but not BO, resulted in a significant increase of GCaMP6m fluorescence intensity in AMB somas (Fig 11A–11D; +CsChrimson, 6.89% ± 1.46% elevation of $\Delta F/F_0$ on average, n = 37; -CsChrimson, 0.21% ± 1.09% elevation of $\Delta F/F_0$ on average, n = 16. Fig 11E–11H; +CsChrimson, 0.98% ± 1.71% elevation of $\Delta F/F_0$ on average, n = 23; -CsChrimson, 0.33% ± 0.86% elevation of $\Delta F/F_0$ on average, n = 22), consistent with the idea that AMBs receive inputs from C4da neurons, but not from BO. We further tested that AMBs indeed relay physiological blue light information from the C4da pathway. To this end, we expressed GCaMP6s in AMBs and irradiated blue lights to activate C4da pathway physiologically. We found that control larvae showed a significant increase in the fluorescence intensity in the AMB soma in response to

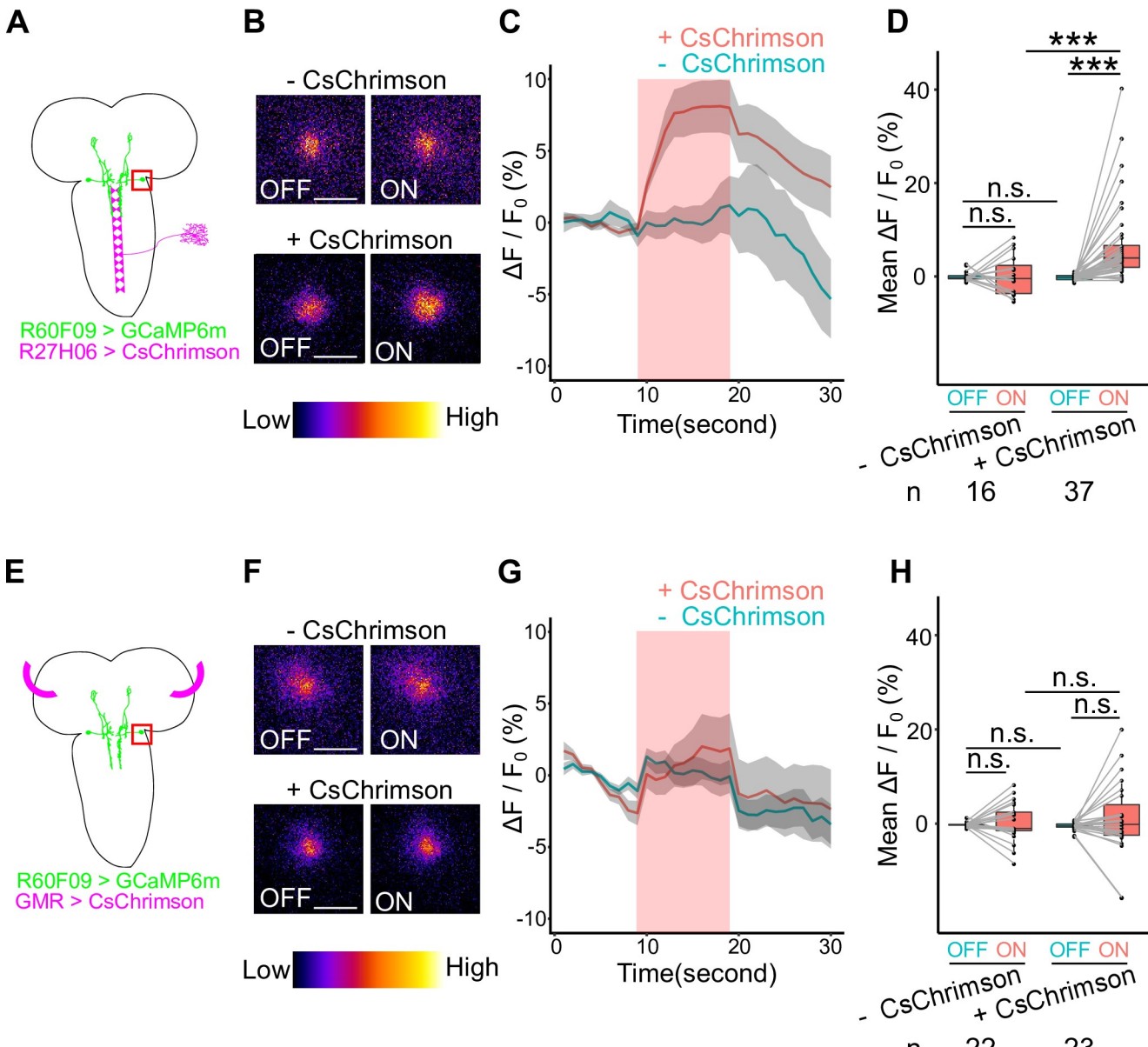

**Fig 11. Optogenetic activation of C4da neurons causes Ca²⁺ responses in AMBs.** (A, E) Schematic views of Ca²⁺ imaging on AMB's soma during optogenetic activation of sensory neurons. Genotypes: *w; UAS-GCaMP6m, tsh-GAL80/R27H06-LexA; R60F09-GAL4/LexAop-CsChrimson* (+*CsChrimson*); *w; UAS-GCaMP6m, tsh-GAL80/+; R60F09-GAL4/LexAop-CsChrimson* (-*CsChrimson*) (A-D); *w; GMR-GAL4/R60F09-LexA; UAS-CsChrimson/LexAop-GCaMP6m* (+*CsChrimson*); *w; R60F09-LexA/+; UAS-CsChrimson/LexAop-GCaMP6m* (-*CsChrimson*) (E-H). (B, F) Ca²⁺ imaging of AMB neurons upon optogenetic activation of sensory neurons in +CsChrimson (lower panels) and -CsChrimson (upper panels) conditions. Here are shown representative images of relative Ca²⁺ levels 5 seconds before (OFF) and after (ON) light application. Scale bars, 10μm. (C, G) Time series of Ca²⁺ responses in the soma of AMBs upon optogenetic activation of sensory neurons. We applied stimulation in the period indicated by the red band. Data are shown as the mean ± SEM. (C) -CsChrimson, n = 16; +CsChrimson, n = 37. (G) -CsChrimson, n = 22; +CsChrimson, n = 23. (D, H) Average of AMB ΔF/F₀ values in 10 seconds before (OFF) or during (ON) optogenetic activation. In the boxplot, the width of the box represents the interquartile range. The whiskers extend to the data point which is less than 1.5 times the length of the box away from the box, and the dot represent outlier. We assessed statistical significance by paired *t*-test for paired samples and Welch's two sample *t*-test for unpaired samples. ***p < 0.001.

blue light irradiation whereas no significant increase of the fluorescence intensity was observed in larvae ablated C4da neurons (S5 Fig; +*rpr*, 0.74% ± 2.23% elevation of ΔF/F0 on average, n = 43; -*rpr*, 7.40% ± 3.17% elevation of ΔF/F0 on average, n = 40). This data indicates that

blue light irradiation activates AMB neurons via C4da neurons. Taken together, our data indicate that AMBs preferentially convey blue light information from C4da sensory neurons to MDNs to evoke backward locomotion.

## AMBs and MDNs are dispensable for dead end-induced backward locomotion in larvae

Previous reports indicate that MDNs are required for the dead end-evoked backward walking in adult flies [6,9]. We therefore examined whether MDNs and AMBs were similarly involved in the dead end-evoked backward responses in larvae using a narrow chamber that limits larval lateral and rotational movements. When effector control larvae encountered the end of the narrow chamber, they tried crawling forward several times and then changed their crawling direction by repeated backward locomotion (Fig 12 and S4 Movie). Compared to control larvae, silencing of either C3da or C4da and BO caused no obvious defect in the dead end-evoked backward locomotion (Fig 12A; Effector control, 5 in the median, n = 20; *ppk-GAL4* ablation, 4 in the median, n = 20; *GMR-GAL4* ablation, 6.5 in the median, n = 20; *ppk-GAL4 + GMR-GAL4* ablation, 4.5 in the median, n = 20; *GAL4*$^{19-12}$ ablation, 7 in the median, n = 20; *NompC-GAL4* ablation 4.5 in the median, n = 20). Similarly, no significant changes in backward locomotion upon reaching the dead ends were observed in larvae expressing TNT in AMBs or MDNs using *AMB-GAL4* or *MDN-FLP-GAL4*, respectively (Fig 12B and S4 Movie; Effector control, 4 in the median, n = 20; *AMB-GAL4* control, 3 in the median, n = 20; *AMB-GAL4* silencing, 3 in the median, n = 20; *MDN-FLP-GAL4* control, 4.5 in the median, n = 20; *MDN-FLP-GAL4* silencing, 4.5 in the median, n = 20). These data suggest that, unlike in adult flies, MDNs are dispensable for the dead end-induced backward locomotion in larvae.

## Discussion

Command-like neurons for backward locomotion have been described in several animal species, including AVA neurons in *C. elegans* and MDNs in *Drosophila* [4,6,7], yet how they are activated by sensory inputs remains to be understood. In this study, we have identified a novel class of ascending interneurons, AMBs, that activate the command-like MDNs to elicit backward locomotion in *Drosophila* larvae. This notion is supported by the following lines of evidence. First, optogenetic activation of AMBs can induce backward locomotion similar to that induced by MDN activation (Fig 1 and S4 Fig). Second, AMB activation induces Ca$^{2+}$ responses in MDNs, consistent with AMB functioning upstream of MDN (Fig 6). Third, AMB-induced backward locomotion is attenuated by silencing of MDNs (Fig 7). TEM-based connectome studies suggested that larval MDNs do not appear to have direct inputs from sensory neurons [7]; our studies suggest that AMBs relay inputs from sensory neurons to MDNs. Indeed, our dual labeling of AMBs and MDNs revealed that MDN dendrites are closely apposed to AMB axon terminals (Fig 4). Furthermore, t-GRASP data suggest that AMB axons likely have direct connections with MDN dendrites (Fig 5). Given that AMBs are cholinergic neurons (Fig 3), it is likely that AMBs are presynaptic neurons that provide excitatory inputs to MDNs (Fig 13). In addition to MDNs, AMBs might have other downstream neurons to induce backward locomotion, as substantial AMB-induced backward locomotion activity remained even in the larvae with silencing MDNs (Fig 7). Future studies using the TEM-based connectome analysis might lead to better understanding of how AMBs contribute to backward locomotion in coordination with other neurons including MDNs.

In theory, animals move backward when they encounter insurmountable obstacles or potentially aversive situations. Indeed, multiple aversive stimulations including mechanical stimulation and blue light irradiation on the head trigger backward locomotion in larvae. Our

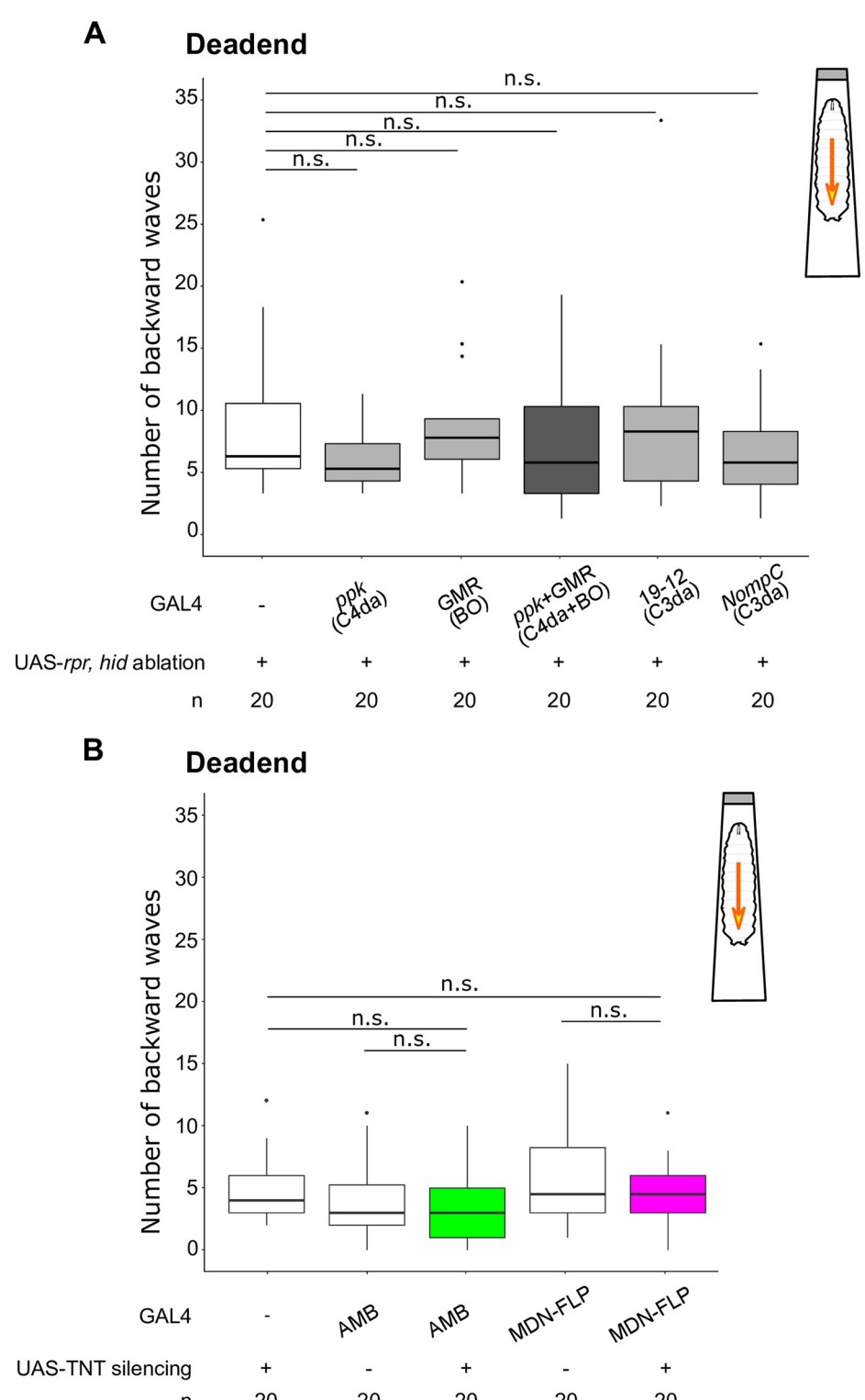

**Fig 12. AMBs and MDNs are dispensable for dead ends-evoked backward waves in larvae.** The number of backward waves evoked by the dead ends with ablation of sensory neurons (A), or silencing AMBs or MDNs (B). The number of backward waves was counted for 1 min after an animal encounter the dead end in the chamber. Genotypes are shown in S1 Table. In the boxplot, the width of the box represents the interquartile range. The whiskers extend to the data point which is less than 1.5 times the length of the box away from the box, and the dot represent outlier. We assessed statistical significance by the Wilcoxon rank sum test with the Holm method.

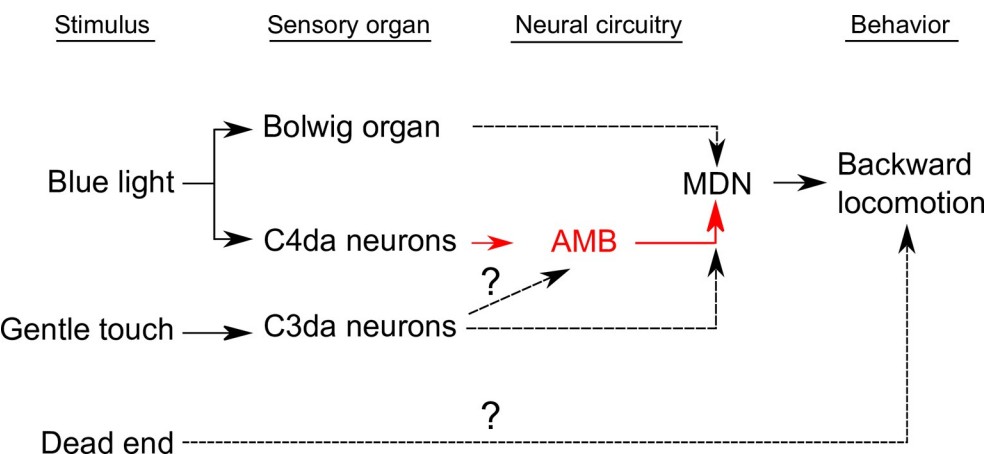

**Fig 13. A circuit diagram of larval sensory systems and their potential downstream neurons that induce backward locomotion.** Black solid lines indicate the functional connections shown in the previous studies[7,14,40]. Red solid lines indicate the functional connections shown in this study. Black dashed lines indicate the functional connections implied in this study.

data suggest that different but partially redundant pathways act downstream of sensory neurons to induce backward locomotion in response to distinct sensory stimuli (Fig 13). Blue light-evoked escape behavior is typically mediated by BO and C4da neurons in larvae [14]. Our data suggest that aversive blue light information from BO and C4da are relayed by the AMB-independent and -dependent pathways, respectively and that both pathways eventually converge on MDNs to evoke backward locomotion (Figs 9, 10 and 13, S5 Fig). Although MDNs are previously shown to be required for backward locomotion in response to mechanical stimuli on the larval heads [7], it was unknown whether MDNs are also required to evoke backward locomotion in response to other aversive sensory modalities. Our data indicate that MDNs are indeed required for backward locomotion in response to aversive blue light irradiation as well as gentle touch (Figs 8 and 9). In larvae, gentle touch and blue lights are typically received by at least three different sensory systems, C3da neurons, BO, and C4da neurons [14,40]. We thus propose that MDNs are a convergence point for multiple aversive sensory inputs to trigger backward locomotion(Fig 13). It is of importance to examine whether other aversive sensory modalities, such as high temperature, high salt and bitter taste, might also require MDNs to evoke escape behavior including backward locomotion. It is also interesting to examine whether AMBs specifically mediate blue light information from C4da neurons or are also recruited by other aversive modalities.

Gentle touch-evoked backward locomotion was largely blocked by MDN silencing, but not by AMB silencing (Fig 8). Given that gentle touch on the head is predominantly mediated by C3da sensory neurons, C3da activity might evoke MDN activation through AMB-independent pathways (Fig 13). An alternative scenario is that, similar to the blue light-evoked backward locomotion, redundant circuits including the AMB-MDN pathway might function downstream of C3da neurons and they might eventually converge on MDNs (Fig 13). C3da neurons are reported to innervate multiple secondary neurons including DP-ilp7 neurons [26], DnB neurons [25], and Wave neuron [13]. In particular, anterior Wave neurons in the ventral nerve cord (VNC) act downstream of C3da/C4da sensory neurons to evoke backward locomotion in response to noxious mechanical stimulation while dispensable for blue light-evoked backward locomotion [13]. It is feasible to examine whether these C3da secondary neurons might function in the circuits that convey mechanical aversive signals from C3da neurons to MDNs.

Unlike mechanical stimuli and blue light irradiation on the head, the dead end-evoked backward locomotion appears to be independent of AMBs and MDNs, as no significant change in backward locomotion was observed by silencing either AMBs and MDNs (Figs 12 and 13). In addition, silencing of either C3da or C4da and BO caused no obvious defects in the dead ends-evoked backward locomotion (Fig 12). This is marked contrast to the backward walk in adult flies, as the dead end-evoked backward walk in adult flies largely depends on MDN activity [6]. It is most likely that adult flies receive mechanosensory information on the anterior legs when they encounter dead ends and that the sensory information inputs MDNs through TwoLumps neurons in VNC to evoke backward walking [9]. By contrast, larvae typically try to crawl forward repeatedly when they encounter the dead ends and subsequently switch to backward locomotion (S4 Movie). These observations imply that repeated mechanosensory inputs on the head as well as the anterior body wall might contribute to induction of backward locomotion. In addition, larvae might have a circuit mechanism to evoke backward locomotion independent of MDNs. Indeed, in the dissected larval VNC without the brain, optogenetic activation of anterior Wave neurons could induce backward $Ca^{2+}$ waves in the motor circuits [13], suggesting that sensory inputs from anterior Wave neurons could evoke backward locomotion independently of the brain circuits including MDNs in larvae. Further studies in the cellular and network levels will be needed to understand how much similar and distinct neural circuits as well as sensory systems might be utilized in adult flies and larvae to evoke backward movements in a context dependent manner.

In summary, by the use of functional optogenetic and *in vivo* imaging techniques in the larval neural circuits, we have revealed circuits by which distinct sensory pathways converge on the command-like neurons to evoke backward locomotion. We also provide a possibility that distinct but partially redundant pathways function to evoke backward locomotion in adult and larvae as well as in response to different aversive stimuli. Given its relative simplicity, combined with the powerful genetic tools in *Drosophila*, further studies of circuit mechanisms of the backward locomotion will help to elucidate how multiple sensory inputs are coordinated in the cellular and circuit level to evoke particular behavior in response to a variety of external cues.

## Materials and methods

### Fly stocks

We used both male and female early third instar larvae (AEL 72–96 h) of *Drosophila melanogaster* in all experiments. We raised larvae on standard medium at 25˚C in total darkness unless otherwise specified. We obtained fly stocks carrying *w1118* (BL#3605), *tub-GAL80ts*[42] (BL#7017), *tub-FRT-GAL80-FRT* (BL#38880), *UAS-TNT* (BL#28838), *UAS-CsChrimson* (BL#55135, BL#55136), *R27H06-LexA* (BL#54751), *R60F09-GAL4* (BL#39255), *R60F09-LexA* (BL#61576), *R73D06-GAL4* (BL#46692), *R73F04-GAL4* (BL#49623), *R60F09-GAL4DBD* (BL#75644), *R11E07-p65AD* (BL#68816), *UAS-mCD8GFP* (BL#5137), *UAS-GCaMP6m* (BL#42748), *LexAop-CsChrimson* (BL#82183), *LexAop-GAL80* (BL#32215), *LexAop-GCaMP6m* (BL#44276), *LexAop-GCaMP6s* (BL#44590), *UAS-mCD8RFP* (BL#32229), *LexAop-mCD8GFP* (BL#32229), *GMR-GAL4* (BL#1104), *ppk-GAL4* [43, 44], *NompC-GAL4* (BL#36369), *UAS-post-t-GRASP and LexAop-pre-t-GRASP* (BL#79039) and *UAS-DenMark* (BL#33061) from Bloomington Drosophila Stock Center. *GAL419-12*, *repo-GAL80* from Jay Parrish (University of Washington); *Rh6-LexA* from Sandra Berger-Müller (Max Planck Institute of Neurobiology); *UAS-rpr*, *UAS-hid* from Douglas Allan (University of British Columbia); *Otd-FLP* from David Anderson (California Institute of Technology); *tsh-GAL80* from Gero Miesenböck (University of Oxford); *UAS-brpD3::mCherry* from Takashi Suzuki (Tokyo

Institute of Technology); *LexAop-rCD2RFP* from Tzumin Lee (Janelia Research Campus); *SS01613-GAL4* from Chris Doe (University of Oregon). *R73F04-LexA* by cloning R73F04 from the genome of *w^{1118}* using the primers described in FlyLight (http://flweb.janelia.org/) into *pENTR 1A* (Thermo Fisher Scientific, A10462) and subsequently to *pBPnlsLexA::p65Uw* (Addgene, #26230) according to the previously described method [45]. We generated *Gad1-2A-GAL80* using CRISPR/Cas9 system as described in the previous studies [46,47]. We inserted a transgene encoding the 2A peptide and *GAL80* gene in front of the stop codon of the *Gad1* gene with the following 20-bp guide RNA (gRNA) sequence: 5′-GCCTGGGCGAC-GACTTGTAA-3′. The engineered locus encodes a bicistronic transcript that produces Gad1 and GAL80 proteins, thereby allowing us to express *GAL80* in the same spatiotemporal pattern as endogenous Gad1. In order to generate the split GAL4 line that specifically label AMBs (*AMB-GAL4*), we screened the Janelia p65AD lines that label SOG neurons by crossing with flies harboring *UAS-CsChrimson* and *R60F09-GAL4DBD*. Among ~50 lines screened, we finally defined *R11E07-p65AD*, as larvae harboring *UAS-CsChrimson*, *R60F09-GAL4DBD*, and *R11E07-p65AD* exhibited robust backward locomotion upon optogenetic activation. Indeed, we found specific labeling of AMBs in larvae harboring *UAS-CsChrimson*, *R60F09-GAL4DBD*, and *R11E07-p65AD* (Fig 1D).

## Optogenetic screen

In order to systematically identify neurons that show a specific type of escape behavior upon activation, we designed an optogenetic screen in third instar *Drosophila* larvae. This screen used the Janelia collection of GAL4 lines [27], *UAS-CsChrimson* [28] and *tsh-GAL80* [48,49] transgenes to express light-gated cation channels in arbitrary neurons locating in the brain or the SEZ. We first selected 783 GAL4 lines which label less than 20 neurons in the hemisphere brain. The larvae grew in standard medium containing 0.5 mM ATR at 25˚C. We floated larvae using 20% sucrose, and then gently washed to collect them on an agarose plate. The behavioral experiment was conducted on a 1% agarose gel plate in a 9 cm plastic dish. We placed 15 larvae on the center of the plate for each trial and performed three trials for each genotype. For optogenetic activation, we applied 617 nm light (Thorlab, M617L4, 34.0 μW/mm$^2$) for 5 minutes. We recorded the larval behavior with a CCD Camera (Thorlab, 1500M-GE) at a capture rate of 1 fps with infrared background illumination (CCS, LDR2-132IR940-LA), and classified the behavior phenotype manually.

## Immunohistochemistry

We dissected early third instar larvae in phosphate buffered saline (PBS) and fixed them in 4% paraformaldehyde/PBS for 15 minutes at room temperature. For imaging without staining, we moved the larval CNS into 0.3% Triton X-100/PBS (PBT) and incubated 3 hours at 4˚C after fixation. Then we moved the samples into VECTASHIELD mounting medium (Vector Laboratories, H-1000) and incubated 3 hours, and imaged with confocal microscopy (Leica TCS SP8). For imaging with immunohistochemistry, we moved the samples into PBT and incubate 30 minutes after fixation and blocked for 30 minutes in PBT containing 5% normal goat serum (NGS) at 4˚C on a shaker. We subsequently incubated the samples with the primary antibody diluted in 5% NGS/PBT at 4˚C overnight. After five times wash with PBT, we blocked samples for 30 minutes in 5% NGS/PBT. We subsequently incubated the samples with the secondary antibody diluted in 5% NGS/PBT at 4˚C overnight. After five 10 minutes washes with PBT, we transferred the stained samples into VECTASHIELD for 3 hours at 4˚C and imaged with confocal microscopy. The list of the antibodies used in this study and the dilution are as follows: anti-ChAT (mouse monoclonal; hybridoma bank 4B1; 1:50), anti-GABA (rabbit

polyclonal; Sigma Aldrich #110M4781; 1:100), anti-VGlut (rabbit polyclonal; a gift from Hermann Aberle [50]; 2-DVl-lut-N-TRIM (N); 1:400), anti-mouse Alexa Fluor 635 (goat IgG; Molecular Probes, #A31575), and anti-rabbit Alexa Fluor 633 (goat IgG; Molecular Probes, #A21071).

## Calcium imaging

Larvae were grown in the standard medium containing 0.5 mM all-trans-retinal (ATR; Sigma Aldrich, #R2500). Third instar larvae were pinned down on a silicon dish (Silpot 184, Dow Corning Toray), and dissected along the dorsal midline in calcium-free HL3.1 buffer (NaCl 70 mM, KCl 5 mM, $MgCl_2$ 4 mM, $NaHCO_3$ 10 mM, Trehalose 5 mM, Sucrose 115 mM, HEPES 5mM, pH 7.2 [51]). We removed the internal organs except for neural tissues. We imaged the CNS using an Olympus BX51WI microscope equipped with a spinning-disk confocal unit Yokogawa CSU10 (Yokogawa) and an EM-CCD digital camera (Evolve, Photometrics). For activation of neurons expressing CsChrimson, we applied 615 nm light ($105\ \mu W/mm^2$) with a pE-100 device (CoolLED). For activation of blue light-sensitive neurons physiologically, we applied 475 nm light ($208\ \mu W/mm^2$) with a pE-300$^{ultra}$ device (CoolLED). We quantified calcium probe signals using Fiji (Fiji is just ImageJ) and R (ver 3.2.2) We set ROIs on the neurites of MDNs or the soma of AMBs, and calculated mean signal intensity within each ROI using Fiji. We measured the raw signal intensity F and calculated $F_0$ as the mean fluorescent signals from 0 to 10 seconds before the onset of optogenetic stimulation, and treated it as a baseline. Then we calculated the normalized calcium transient in each ROI according to the formula $\Delta F/F = (F-F_0)/F_0$ using R. We calculated mean $\Delta F/F$ in ON or OFF state as the mean of $\Delta F/F$ in 10 seconds after or before the onset of optogenetic stimulation.

## Optogenetics in free-moving larvae

Larvae were grown in standard medium containing 0.5 mM ATR at 25˚C except for the R60F09-Brain experiments in Fig 1. For R60F9-Brain, we grew larvae 3 days at 18˚C and 1 day at 29˚C. We floated larvae using 20% sucrose, and then gently washed to collect them on an agarose plate. The behavioral experiment was conducted on a 1% agarose S gel plate (Wako, #13–90231) in a 90 mm plastic dish. We placed one larva on the center of the plate at one trial. For optogenetic activation, we applied 30 seconds of 640 nm light (Lumencor Spectra X7, $93.3\ \mu W/mm^2$ for Figs 1 and 10 and S2 Fig; $2.84\ \mu W/mm^2$ for Fig 7) for CsChrimson activation. We recorded the larval behavior with a sCMOS-Camera (Andor, Zyla 5.5) at frame rate of 20 fps under a stereomicroscope (Olympus, MVX10) with infrared background illumination (CCS, LDR2-132IR850-LA).

## Blue light assay

We prepared the larvae and equipment in the same way as optogenetics assay above. For stimulation, we applied 440 nm spotlight ($0.5\ cm$, $245\ \mu W/mm^2$, Lumencor Spectra X7) to the larvae on the agarose plate for five seconds. We targeted the light application to the anterior half of the larval body in order to minimize the variance of the stimulation. If the light did not cover the anterior half of the body, we excluded the trial from the analysis. We performed one trial for each animal.

## Gentle touch assay

We prepared the larvae and equipment in the same way as described in the blue light assay above. As a stimulus, we used an eyelash hair that was glued to the end of a pipette tip which

could apply a force in the range of 1–10 μN. We stroked the animal four times with intervals of 15 seconds in 1 minute for one trial. We performed one trial for each animal.

### Dead end assay

We prepared the larvae and equipment in the same way as described in the blue light assay above. We cut the terminal 1cm segment off of a 200 μl pipette tip (WATSON) and heat-sealed the tip to make a "dead-end" chamber that limits larval lateral and rotational movement. We introduced an individual larva into the chamber and waited until it physically encountered the end of the narrow chamber. We started recording for 1 minute from their encountering the dead end. When the larva failed to reach the dead end within 1 minute from the entry to the tip, we excluded the trial from the analysis. We performed one trial for each animal.

### Behavioral analysis

We quantified larval forward/backward waves manually. We counted larval waves when larvae showed a sequence of muscle contractions across segments directed from anterior to posterior (backward) or posterior to anterior (forward). For making behavior ethograms and calculating time spent in a behavioral mode, we utilized FIMTrack to track larval behavior. We picked up "go state" and "bending state" among the parameters calculated by the software. We defined "forward" as the state in which "go state" is on and backward waves have not occurred, "backward" as the state in which "go state" is on and backward waves have occurred, "backward bend" as the state in which "bending state" is on and backward waves have occurred, "bend" as the state where either right or left "bending state" is on, "stop" as the state where both "go state" and "bending state" are off.

### Statistical analysis

We evaluated statistical significance using the Wilcoxon rank sum test, Fisher's exact test or Welch's t test. Asterisks denote statistical significance: *** $p < 0.001$; ** $p < 0.01$; * $p < 0.05$; n. s., not significant. Error bars represent standard errors of the mean (SEM). All of the statistical analysis were performed by R version 3.3.2. We did not use any methods to determine whether the data met assumptions of the statistical approach.

### Data and software availability

The datasets generated and/or analyzed during the study are available from the corresponding author on reasonable request.

## Supporting information

**S1 Fig. R73F04-GAL4 labels MDNs.** (A) Expression pattern of *MDN-labeling GAL4* used in this study. The yellow arrowheads indicate the soma of MDNs. Genotypes: *w*; *UAS-mCD8GFP, tsh-GAL80/+*; *R73F04-GAL4/+(R73F04-GAL4)*; *w*; *UAS-mCD8GFP, tsh-GAL80/+*; *R73F04-GAL4, Gad1-2A-GAL80/+* (*MDN-ACh*); *w*; *UAS-mCD8GFP/Otd-FLP, tub-FRT-GAL80-FRT*; *R73F04-GAL4, Gad1-2A-GAL80/+* (*MDN-FLP*). Scale bar, 100 μm. (B) Dual-labeling with *MDN-labeling GAL4* used in the previous study [7] and *R73F04-LexA*. The yellow dot square in the upper row indicates the area shown in the lower row. The yellow arrowheads indicate both *R73F04-LexA* and *SS01613-GAL4* label the soma of MDNs. Scale bar, 50 μm. (TIF)

**S2 Fig.** Two distinct GAL4 lines label AMBs (A) Expression patterns of R60F09-LexA in the third instar larval CNS. The yellow arrowheads indicate the soma of AMBs. Maximum

intensity projection of the entire CNS shown. Genotypes: LexAop-mCD8GFP, UAS-mCD8RFP/+; R60F09-LexA/+; +/+. Scale bar, 100 μm. (B, C) The number of backward/forward waves in 10 seconds before (OFF) or during (ON) optogenetic activation with CsChrimson. n = 15 for each genotype. We assessed statistical significance by the Wilcoxon rank sum test with Holm method. ***p < 0.001. (D) R60F09-LexA and R73D06-GAL4 co-label AMBs. The yellow dot square indicates the area shown in the lower row. The yellow arrowheads indicate the somas of AMBs. Genotypes: LexAop-mCD8GFP, UAS-mCD8RFP/+; R60F09-LexA/+; R73D06-GAL4/+. Scale bars, 50 μm. (E) GAL80 labeled by R60F09-LexA diminishes CsChrimson expression in AMBs labeled by R73D06-GAL4 co-labeling. The yellow arrowheads indicate the somas of AMBs. Genotypes: w; LexAop-rCD2RFP/R60F09-LexA, tsh-GAL80; R73D06-GAL4/UAS-CsChrimson (the upper row), w; LexAop-rCD2RFP/R60F09-LexA, tsh-GAL80; R73D06-GAL4/UAS-CsChrimson, LexAop-GAL80 (the lower row). Scale bars, 50 μm. (F, G) The number of backward/forward waves in 10 seconds before (OFF) or during (ON) optogenetic activation with CsChrimson. n = 18, 20 for each genotype. In the boxplot, the width of the box represents the interquartile range. The whiskers extend to the data point which is less than 1.5 times the length of the box away from the box, and the dot represent outlier. We assessed statistical significance by the Wilcoxon rank sum test with Holm method. ***p < 0.001. (H) Probability of animals showing rolling behavior in 10 seconds before (OFF) or during (ON) optogenetic activation with CsChrimson. We assessed statistical significance by the Fisher's exact test.
(TIF)

**S3 Fig. Positive and negative controls for anti-ChAT immunohistochemistry.** (A) Positive control for anti-ChAT. Anti-ChAT stained MDN somas that are reported to be cholinergic neurons in the previous study [7]. (B) Negative control of 2nd antibodies for immunohistochemistry. No detectable immunostaining was observed in the AMB somas without anti-ChAT antibody. Scale bars, 10 μm.
(TIF)

**S4 Fig. Optogenetic activation of AMBs or MDNs triggers similar repetitive backward locomotion.** (A) Behavior events are color-coded: forward movement (grey), stop (white), bending (yellow), bending with backward locomotion (orange), and backward locomotion (red). (B-D) Behavior ethograms upon optogenetic stimulation of AMBs or MDNs. An animal expressing CsChrimson in either population was subjected to optogenetic activation for 30 seconds. Representative data from 10 different animals are shown for each genotype. (E-H) The number of backward/forward waves or percentage of time spent in a behavioral mode in 10 seconds before (OFF) and during (ON) optogenetic AMB activation with CsChrimson while silencing MDNs. In the boxplot, the width of the box represents the interquartile range. The whiskers extend to the data point which is less than 1.5 times the length of the box away from the box, and the dot represent outlier. We assessed the statistical significance by the Wilcoxon rank-sum test with the Holm method. ***p < 0.001.
(TIF)

**S5 Fig. Blue light irradiation evokes Ca$^{2+}$ responses in AMBs via C4da neurons.** (A) A schematic view of Ca$^{2+}$ imaging on AMB's soma while ablating C4da neurons. Genotypes: *UAS-rpr/+; R60F09-LexA, LexAop-GCaMP6s; ppk-GAL4/+ (+rpr); UAS-rpr/+; R60F09-LexA, LexAop-GCaMP6s; +/+ (-rpr)*. (B) Time series of Ca$^{2+}$ responses in the soma of AMBs upon blue light irradiation. We applied stimulation in the period indicated by the blue band. Data are shown as the mean ± SEM. *-rpr*, n = 40; *+rpr*, n = 43. (C) Average of AMB $\Delta F/F_0$ values in 5 seconds before (OFF) or last 5 seconds during (ON) optogenetic activation. In the boxplot, the

width of the box represents the interquartile range. The whiskers extend to the data point which is less than 1.5 times the length of the box away from the box, and the dot represent outlier. We assessed statistical significance by paired t test for paired samples and Welch's two sample *t*-test for unpaired samples. $^{**}$p $<$ 0.01.
(TIF)

**S1 Movie. Activation of AMBs or MDNs evokes backward locomotion.** The movie shows the response of effector control, AMB $>$ CsChrimson, and MDN-FLP $>$ CsChrimson larvae before and after optogenetic activation of the neurons. The movie is played at 2 times speed.
(MP4)

**S2 Movie. Application of gentle touch evokes backward locomotion.** The movie shows the response of effector control, AMB $>$ TNT, and MDN-FLP $>$ TNT larvae toward gentle touch by an eyelash.
(MP4)

**S3 Movie. Application of a blue light spot evokes backward locomotion.** The movie shows the response of effector control, AMB $>$ TNT, and MDN-FLP $>$ TNT larvae upon 5 seconds of blue light stimulation. The white circle in the movie indicates the area of a blue light spot.
(MP4)

**S4 Movie. Dead ends evoke backward locomotion.** The movie shows the behavior of effector control, AMB $>$ TNT, and MDN-FLP $>$ TNT larvae in the dead-end narrow chamber. All the larvae shown here are headed to left side.
(MP4)

**S1 Table. Genotypes of GAL4 lines utilized in Fig 1.**
(XLSX)

**S2 Table. Numerical data and statistical analysis used in this study.**
(XLSX)

## Acknowledgments

We would like to thank C. Doe, D. Anderson, D. Allan, G. Miesenböck, H. Herranz, P. Soba, S. Berger-Müller, T. Suzuki, T. Lee, Bloomington Stock Center for reagents; FIM-Team for helping data analysis; S. Kondo for helpful advice in vector construction; the members of Emoto lab for critical comments and discussion; M. Miyahara and H. Itoh for technical assistance; J. Parrish for critical reading and comments; the Graduate Program for Leaders in Life Innovation (GPLLI) for supporting N.O-I.

## Author Contributions

**Conceptualization:** Natsuko Omamiuda-Ishikawa, Kazuo Emoto.

**Data curation:** Natsuko Omamiuda-Ishikawa.

**Formal analysis:** Natsuko Omamiuda-Ishikawa.

**Funding acquisition:** Natsuko Omamiuda-Ishikawa, Kazuo Emoto.

**Investigation:** Natsuko Omamiuda-Ishikawa, Moeka Sakai.

**Methodology:** Natsuko Omamiuda-Ishikawa.

**Project administration:** Natsuko Omamiuda-Ishikawa, Kazuo Emoto.

**Resources:** Natsuko Omamiuda-Ishikawa, Moeka Sakai, Kazuo Emoto.

**Software:** Natsuko Omamiuda-Ishikawa.

**Supervision:** Kazuo Emoto.

**Validation:** Natsuko Omamiuda-Ishikawa.

**Visualization:** Natsuko Omamiuda-Ishikawa.

**Writing – original draft:** Natsuko Omamiuda-Ishikawa, Kazuo Emoto.

**Writing – review & editing:** Natsuko Omamiuda-Ishikawa, Kazuo Emoto.

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
