## [Decision Letter · Decision Letter 0]

12 May 2020

Dear Kazuo,

Thank you very much for submitting your Research Article entitled 'A pair of ascending neurons in the subesophageal zone mediates aversive sensory inputs-evoked backward locomotion in Drosophila larvae' to PLOS Genetics. Thank you, also, for your patience in these times of international crisis, which unfortunately affects us all in our reviewing and editing activities.

Your paper was reviewed by three experts, which all agreed that the submitted work is of high interest and technical quality. However, they also raised substantial concerns that need to be addressed. In particular, two referees raised major points whether mechanical touch or light are indeed activating AMB neurons. As pointed out by the reviewers, the current data does not fully support the presented model in Fig. 13. This should ideally be addressed experimentally and should take into consideration previous work, e.g. that class IV da neuron-dependent harsh touch on the head also induces backward locomotion (Takagi et al. Neuron 2017).

In general, figure labeling and presentation should also be adjusted as proposed by the reviewers to make the data more accessible and appreciable, e.g. adjustment of brightness of immunohistochemical signals.

Based on the reviews, we will not be able to accept this version of the manuscript, but we would be willing to review again a much-revised version. We cannot, of course, promise publication at that time.

If you decide to revise the manuscript for further consideration at PLOS Genetics, please aim to resubmit within the next 120 days, unless it will take extra time to address the concerns of the reviewers, in which case we would appreciate an expected resubmission date by email to plosgenetics@plos.org.

[LINK]

We hope that you find the reviewers’ comments helpful in preparing your revised manuscript.

Yours sincerely,

Peter Soba, Ph.D.

Guest Editor

PLOS Genetics

Gregory Barsh

Editor-in-Chief

PLOS Genetics

**Comments to the Authors:**

Reviewer #1: In this study the authors identify the sensory circuitry that input to the command neuron for backward movement in Drosophila larvae. Through an optogenetic screen of a collection of GAL4 lines, the authors found that the activation of a pair of neurons in the subesophageal zone, termed AMBs, elicit backward crawling in larvae. They then carefully analyzed the pre- and post-synaptic partners of AMBs, and discovered that AMBs referentially pass sensory inputs from the C4da nociceptors to the command neurons for backward locomotion. Interestingly, this sensory pathway is different from that for backward locomotion elicited by the light-sensing Bolwig’s organ, which does not involve AMBs.

Recent studies have identified the MDNs as the command neurons for backward movement in adult and larval Drosophila. However, how sensory inputs are conveyed to MDNs is unknown. This study fills this gap and is thus an importance contribution to the understanding of sensory processing of aversive cues that elicit backward movements.

The authors applied a battery of approaches in this solid study, including genetic manipulations, t-GRASP, calcium imaging and behavioral analyses. Their savvy use of genetic tricks for identifying and confirming neurons is impressive. Overall, this is an excellent work.

Major concerns:

1) Fig. 9A shows that ablation of either C3da (GAL4[19-12]) or BO (GMR-GAL4), but not that of C4da, reduced backward locomotion elicited by blue light, indicating a possible involvement of C3da and BO. The ablation of a combination of C4da and BO dramatically reduced backward locomotion. I wonder what would happen if they ablate a combination of C3da and BO. This result is important for the model shown in Fig. 13. The authors may also want to perform Ca2+ imaging to determine whether stimulation of C3da activates AMBs.

2) Brp signals shown in Fig. 2B is so dim that it is hard to see them in the upper panel (the low magnification image). Moreover, it would be more convincing if an image showing the lack of (or much less) Brp signals in dendrites is presented for comparison.

3) Fig.3: need negative and positive controls of the immunostaining.

Minor concerns:

1) What are the criteria for selecting the 783 GAL4 lines for the optogenetic screen? Such information is important for readers to assess the scope of the screen.

2) Explain in the legends for Figure 1 what “OFF” and “ON” mean. Also, indicate what the black dots in the box plots mean (outliers?).

3) Fig 6D, 11D&H: avoid bar charts as they do not show data distribution. Use dot plots or box plots, instead.

4) Page 10, “We genetically ablated C3da neurons by expressing the pro-apoptotic genes reaper (rpr) and head involution defective (hid) using the C3da neuron-specific GAL4 drivers

GAL419-12 and NompC-GAL4”. NompC-GAL4 is also expressed in chordotonal neurons.

5) S1 Table (Genotypes of GAL4 lines utilized in Fig 1.): The rationale for including tub-GAL80ts in the flies for cell-type-specific manipulation of neurons for optogenetic experiments is not explained.

Reviewer #2: In this manuscript, the authors present a thorough analysis of neural circuit elements that mediate aversive stimuli-evoked backward locomotion. They focus in particular on a group of ascending neurons, which they named AMBs. They show that optogenetic activation of AMB neurons is sufficient to trigger backward locomotion. They demonstrate that AMB neurons are upstream of the previously defined backward locomotion triggering MDN descending neurons by successfully combining state-of-the art genetic labeling techniques with optogenetic activation and functional imaging experiments. The functional epistasis experiments are carefully performed to reveal that AMBs evokes backward locomotion -at least partially- through MDN neurons. The authors perform a set of ablation/silencing experiments to show that c4da and AMB neurons are dispensable for gentle touch evoked backward locomotion. Then, they show that Bolwig Organs (BOs) and c4da neurons are redundant for blue light irradiation-evoked backward locomotion. They perform (1) critical functional epistasis experiments and; (2) calcium imaging of AMB neurons combined with optogenetic activation of either c4dA or BO neurons to demonstrate that AMB neurons mediate blue light irradiation-evoked backward locomotion by receiving inputs from c4da neurons but not from BOs. They present convincing evidence that MDNs are necessary for both gentle-touch evoked and blue light irradiation-evoked backward crawling. The authors also demonstrate that AMBs and MDNs are dispensable for dead end induced backward locomotion. The manuscript goes beyond the previous findings showing that MDN triggers backward locomotion by dissecting how different sensory stimuli can mediate backward locomotion through MDN-dependent and independent pathways in the Drosophila larva. The findings further highlight that (1) Drosophila larvae utilize partially redundant neural elements to mediate blue light irradiation-evoked backward crawling and (2) distinct aversive sensory modalities can utilize divergent neural pathways to mediate backward locomotion. These findings will be of wide interest as they form the basis to dissect how multiple sensory inputs are integrated to coordinate escape behavior in complex sensory environments. The manuscript elegantly makes use of a range of currently available neurogenetic and opto-physiology tools to perform complex circuit analysis.

Major points:

1- Figure 11A-D shows that AMB neurons respond to C4dA activation. As C4dA neurons are multimodal, this finding does not mean that AMB neurons respond to blue light. The authors should demonstrate that AMB neurons respond to blue light irradiation.

2- In Figure 1E and F authors show how the number of backward and forward waves are affected upon AMB activation and Figure S1B-D show the related ethograms. A detailed analysis of the kinematic variables such as translational velocity and head angular speed before, during and after the stimuli would be a good addition to assess the behavioral phenotype induced by AMB activation in more detail. The authors should also present how other behavioral modes such as head casts and turns are affected upon AMB activation to understand whether AMB activation is specifically involved in backward locomotion.

3- The activation phenotypes of AMBs and MDNs with are represented with ethograms. A detailed analysis of kinematic variables (i.e. angular velocity and translational velocity) with quantification of behavioral states (duration/distance of forward runs and backward crawling, probability of stopping and reorientation) would reveal possible differences between AMB and MDN activation phenotypes.

Minor points:

1- The authors should present the previous findings on the Wave neurons described by Takagi et al. in the introduction part with more detail since they can also trigger backward locomotion in response to mechanical stimulus to the anterior part of the larva.

2- The authors should show the expression pattern of 73D06-Gal4>Chrimson; 60F09-lexA>lexAOp-Gal80 in Figure S2 because lexAOp-Gal80 expression pattern might not recapitulate the expression pattern of lexAOp-mCD8GFP.

3- In Figure S2F, the number of forward waves are significantly reduced even upon Gal80 expression. What is the behavioral phenotype observed in these larvae? The authors should present a video or quantification of other behavioral parameters to explain this observation. The authors should also discuss whether this is due to the expression of CsChrimson in neurons that are not covered by lexAOp-Gal80 (see previous comment).

4- Previous studies suggest that MDNs inhibit forward locomotion by activating an SEZ descending neuron (Carreira-Rosario et al., 2018, Tastekin et al., 2018). Interestingly, the ethogram in Figure S3D shows that MDN-Flp>CsChrimson activation lead to a direct switch from forward runs to backward locomotion. Quantification of wave propagations would contribute to the mechanistic insight of how AMB-induced backward waves are generated by showing whether forward locomotor waves are inhibited without reaching the most anterior segments or backward locomotor waves start after finishing the forward waves.

5- In Figure 7C, AMB activation upon MDN silencing is still able to suppress the forward locomotion raising the question whether AMBs are involved in the inhibition of forward locomotion through previous defined SEZ descending neuron (or other neurons) in an MDN-independent manner. The authors should perform similar functional epistasis and optogenetic activation-functional imaging experiments with AMB and Pair 1/SEZ-DN1 neurons (Carreira-Rosario et al., 2018, Tastekin et al., 2018) to test this possibility.

6- The authors should quantify other behavioral modes such as stopping and turning presented in the ethograms in Figure 7D and E to compare AMB activation alone to AMB activation with MDN silencing.

7- On page 14 wrong reference. “…Previous studies indicate that two different stimuli can evoke backward locomotion in larvae: mechanical stimuli [14] and blue light irradiation [24] on the head…” Reference 24 does not show that blue light irradiation can evoke backward locomotion.

8- Supplementary Videos (except Supplementary Video 1) are mislabeled.

9- On page 14, “…Next, we examined whether eyelash-triggered backward responses are predominantly mediated by the gentle touch-responsive class III da (C3da) mechanosensory neurons…”. The authors should cite Takagi et al., 2017 who showed that C3da are gentle touch responsive.

10- On page 14, the authors should cite Yan et al., 2012 who showed NoMPC-Gal4 labels C3da.

11- The data that show AMBs are dispensable for gentle touch-evoked backward locomotion is convincing. However, AMBs can still receive inputs from C3da neurons, which could be important for the integration of multisensory aversive inputs (Ohyama et al., 2015). The authors could test this by measuring AMB responses upon C3da activation.

12- Although not statistically significant, ablating C3da neurons also lead to a decrease in the probability of backward locomotion upon blue light irradiation. Xiang et al, 2010 have shown that blue light does not evoke activity in C3da neurons. This leads to the question whether the decrease in backward crawling is due to lack of some proprioceptive feedback which might be provided by C3da neurons. The authors should address whether uncoordinated backward locomotion, which might highlight a proprioceptive defect takes place in C3da ablated larvae.

13- The authors quantify the number of backward waves in gentle touch experiments whereas they quantify the probability of backward locomotion in blue light irradiation experiments. The authors should justify the use of different metrics to quantify backward locomotion in these experiments.

14- On page 18. The reader is referred to S6 Movie but there is no S6 Movie in the supplementary data.

15- In figure legends: Wilcox rank sum > Wilcoxon rank sum.

16- Whenever a boxplot is presented, what box boundaries and whiskers represent should be stated.

17- On page 21, “…Gentle touch-evoked backward locomotion was largely blocked by MDN

silencing, but not by AMB silencing (Fig 9)…” Fig 9 should be Fig 8.

Reviewer #3: The authors in this paper address an exciting key question that directly follows from recent work in the field. The authors ask what are the upstream neural networks that relay sensory cues to Mooncrawler Descending Neurons (MDNs) in Drosophila larvae? Previous work found that MDNs have command-like ability to induce backward locomotion and key downstream components of MDN containing circuits had been characterized (Carreira-Rosario et al., 2018). More recently, in adult Drosophila adults, ascending neurons (TwoLumps) was discovered to mediate touch-evoked backward walk through MDN activation in response to touch stimuli on anterior legs (Sen et al., 2019). Understanding what mediates sensory cues through MDNs in Drosophila larvae advances our overall understanding behind the structure of a motor circuit that has been shown to retain key neural components (MDNs) between the larvae and adulthood. The author’s key finding is the discovery of a pair of ascending neurons, AMBs, and convincingly show that AMBs are upstream of MDNs and mediate blue light-evoked backward movement through MDN activation in response to stimulation of C4da sensory neurons. In addition, the author’s nicely showed that the Bolwig’s organ do not connect to the MDNs via the AMBs. The authors also find that C3da sensory neurons are upstream to MDNs but do not show if these neurons are upstream to AMBs. Both of these findings are intriguing and will certainly prompt future studies. Overall, the work presented in this study is convincing, novel, and advances the field, making this paper worthy of publication in PLOS Genetics.

Comments:

At the end of this paper, the authors investigate the role MDNs or any of the upstream neurons identified in this paper are indispensable for dead end-evoked backward movement in larvae. Previous work found MDNs were required for this response in adults. The authors find that MDNs are dispensable for this behavior in larvae. When watching the video of a larvae in a dead-end environment, it appears as though the larvae is scrunched up and then recoils. Potentially this spring-like action drives backward movement. Since adult fly bodies are very different than larval bodies, it isn’t necessarily surprising that different circuits control backward movement in response to this stimuli. It is unclear why the authors included this finding in this paper. Since these findings are not central to the paper, the authors should think about removing this work and saving it for a future publication investigating the neural networks underlying dead end-evoked backward movement in larvae.

ATR controls are commonly used in the field yet the researchers did not, is there a clear rationale behind omitting this standard control?

In the main text, the paragraph that starts on page 12 describes optogenetic activation experiments but no figure reference is given. Could the authors link this text to a figure?

The Figures are not easily accessible to readers outside the field. Please find specific examples below where the authors could make their Figures more accessible.

1) Fluorescence in images could be brighter, see Figure 2. Image brightness can be adjusted in FIJI.

2) While some of the Figures are nicely labeled (Figure 1-5), later figures have poorly labeled schematics and genotype names. For example, in Figure 8, there is a lovely schematic but it isn’t clear what is happening. If there was a header “gentle mechanical touch” and the eyelash probe was labeled, that would greatly aid the reader. Glancing at Figure 8A, unless readers know what ppk, 19-12, and NompC are, it isn’t obvious that these are drivers expressed in either C4da or C3da sensory neurons. Also rpr hid and TNT labels could additionally have a single word describing what is happening –‘ablation’, ‘silencing’. In Figure B, the only GAL4 line that doesn’t need to be searched for in the figure legend, text, or Figure 1 is MDN-FLP. In Figure 10 the y-axis label is ‘Probability (%)’, probability of what? Crawling backward? In Figure 10 in the nerve cord schematics there are red objects that have never been seen before, it would aid the reader if a meaningful label was added.

3) Figure 6A. In the schematic the green tracing looks like AMB yet R60F09-GAL4 is used instead of AMB-GAL4, is this a typo or is R60F09 also AMB-GAL4?

**Have all data underlying the figures and results presented in the manuscript been provided?**

Reviewer #1: Yes

Reviewer #2: No: The authors state that the data sets will be available upon 'reasonable' request from the corresponding author. They should make all the data available in public repositories that allows reanalysis and replication of findings including behavioral and functional imaging experiments.

Reviewer #3: Yes

PLOS authors have the option to publish the peer review history of their article (what does this mean?). If published, this will include your full peer review and any attached files.

Reviewer #1: No

Reviewer #2: No

Reviewer #3: No

---

## [Editor Report · Decision Letter 1]

15 Sep 2020

Dear Kazuo,

We are pleased to inform you that your manuscript entitled "A pair of ascending neurons in the subesophageal zone mediates aversive sensory inputs-evoked backward locomotion in Drosophila larvae" has been editorially accepted for publication in PLOS Genetics. Congratulations!

Yours sincerely,

Peter Soba, Ph.D.

Guest Editor

PLOS Genetics

Gregory Barsh

Editor-in-Chief

PLOS Genetics

Comments from the reviewers (if applicable):

Dear Kazuo,

thank you very much for submitting the revised version of your manuscript. I think the new data and modifications satisfy all major comments of the reviewers and have significantly improved the paper. The comments that were not experimentally addressed are not crucial for the major conclusion of the paper and beyond the scope of the current manuscript. Therefore, we are happy to accept your paper editorially in its current format.

Best,

Peter

**Data Deposition**

http://datadryad.org/submit?journalID=pgenetics&manu=PGENETICS-D-20-00430R1

**Press Queries**

---

## [Editor Report · Acceptance letter]

13 Oct 2020

PGENETICS-D-20-00430R1 

A pair of ascending neurons in the subesophageal zone mediates aversive sensory inputs-evoked backward locomotion in Drosophila larvae 

Dear Dr Emoto, 

We are pleased to inform you that your manuscript entitled "A pair of ascending neurons in the subesophageal zone mediates aversive sensory inputs-evoked backward locomotion in Drosophila larvae" has been formally accepted for publication in PLOS Genetics! Your manuscript is now with our production department and you will be notified of the publication date in due course.

With kind regards,

Jason Norris

PLOS Genetics

On behalf of:
